# A Prototype-Oriented Framework for Unsupervised Domain Adaptation

**Korawat Tanwisuth**[*,1]**, Xinjie Fan**[*,1]**, Huangjie Zheng**[*,1]**, Shujian Zhang**[1]**,**
**Hao Zhang**[2]**, Bo Chen**[3]**, Mingyuan Zhou**[1]
[1]The University of Texas at Austin    [2] Cornell University    [3]Xidian University
`korawat.tanwisuth@utexas.edu, mingyuan.zhou@mccombs.utexas.edu`

## Abstract

Existing methods for unsupervised domain adaptation often rely on minimizing some statistical distance between the source and target samples in the latent space. To avoid the sampling variability, class imbalance, and data-privacy concerns that often plague these methods, we instead provide a memory and computation-efficient probabilistic framework to extract class prototypes and align the target features with them. We demonstrate the general applicability of our method on a wide range of scenarios, including single-source, multi-source, class-imbalance, and source-private domain adaptation. Requiring no additional model parameters and having a moderate increase in computation over the source model alone, the proposed method achieves competitive performance with state-of-the-art methods.

## 1 Introduction

In many real-world applications, such as healthcare and autonomous driving, data labeling can be expensive and time-consuming. To make predictions on a new unlabeled dataset, one may naively use an existing supervised model trained on a large labeled dataset. However, even subtle changes in the data-collection conditions, such as lighting or background for natural images, can cause a model's performance to degrade drastically [1]. This shift in the input data distribution is referred to in the literature as covariate shift [2]. By leveraging the labeled samples from the source domain and unlabeled samples from the target domain, unsupervised domain adaptation aims to overcome this issue, making the learned model generalize well in the target domain [3].

Ben-David et al. [4, 5] provide an $\mathcal{H}$-divergence based theoretical upper bound on the target error. Ganin [6] popularizes learning an invariant representation between the source and target domains to minimize this divergence. Numerous prior methods [7–12] follow this trend, focusing on using the source and target samples for feature alignment in the latent space. While this approach can reduce the discrepancy between domains, directly using the source and target samples for feature alignment has the following problems. First, several commonly used methods that can be used to quantify the difference between two empirical distributions, such as maximum mean discrepancy (MMD) [13] and Wasserstein distance [14, 15], are sensitive to outlier samples in a mini-batch when used to match the source and target marginal distributions [16, 17]. We attribute this problem to the sampling variability of both the source and target samples. Second, while we typically assume that the two domains share the same label space, we cannot guarantee that the samples drawn from the source and target domains will cover the same set of classes in each mini-batch. Especially, if the label proportions shift between domains, learning domain invariant representation might not lead to improvements over using the source data alone to train the model [18]. If we pull the support of

---

* Equal contribution. Corresponding to: `mingyuan.zhou@mccombs.utexas.edu`
PyTorch code is available at `https://github.com/korawat-tanwisuth/Proto_DA`

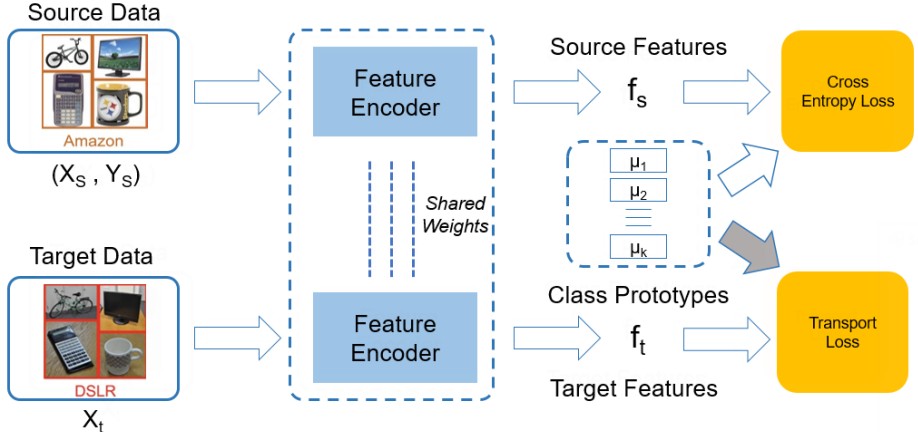

Figure 1: This figure exhibits a diagram of Prototype-oriented Conditional Transport (PCT). Unlike existing methods that align the target and source features, our method aligns the target features with class prototypes. The gray arrow indicates that the gradients of the prototypes do not back-propagate through the transport loss.

the source and target feature representations from different classes closer together, the classifier will be more likely to misclassify those examples. Finally, aligning the target features to source features means that we need access to both the source and target data simultaneously. In applications such as personal healthcare, we may not have access to the source data directly during the adaptation stage; instead, we may only be given access to the target data and the model trained on the source data.

We propose an algorithm that constructs class prototypes to represent the source domain samples in the latent space. Using the prototypes instead of source features avoids the previously mentioned problems: **1)** sampling variability in the source domain, **2)** instance class-mismatching in a mini-batch, and **3)** source-data privacy concerns. As we expect the classifier to make better predictions on the target data in regions where the source density is sufficiently high [19], it is natural to consider encouraging the feature encoder to map the target data close to these prototypes. Motivated by the cluster assumption [20] (decision boundaries should not cross high-density regions of the data), we provide a method to transport the target features to these class prototypes and vice versa. We further extend our bi-directional transport to address the potential shift in label proportions, a common problem that has been studied [21–25] but that has been often overlooked in prior works [6, 7, 26].

Compared to existing methods, the proposed one has several appealing aspects. First, it does not rely on adversarial training to achieve competitive performance, making the algorithm robust and converge much faster. Moreover, learnable prototypes not only avoid expensive computation but also bypass the need to directly access the source data. This attribute makes our algorithm applicable to the settings where preserving the source data privacy is a major concern. Unlike clustering-based approaches that typically require multiple forward passes before an update, our algorithm processes data in mini-batches for each update and is trained in an end-to-end manner.

We highlight the main contributions of the paper as follows: **1)** We utilize the linear classifier's weights as class prototypes and propose a general probabilistic framework to align the target features to these prototypes. **2)** We introduce the minimization of the expected cost of a probabilistic bi-directional transport for feature alignment, and illustrate its superior performance over related methods. **3)** We test the proposed method under multiple challenging yet practical settings: single-source, multi-source, class-imbalance, and source-private domain adaptation. In comparison to state-of-the-art algorithms, the proposed prototype-oriented method achieves highly competitive performance in domain adaptation, while requiring no additional model parameters and only having a moderate increase in computation over the source model alone.

## 2    Prototype-oriented conditional transport

In this section, we propose Prototype-oriented Conditional Transport (PCT), a holistic method for domain adaption consisting of three parts: learning class prototypes, aligning the target features with learned prototypes using a probabilistic bi-directional transport framework of Zheng and Zhou

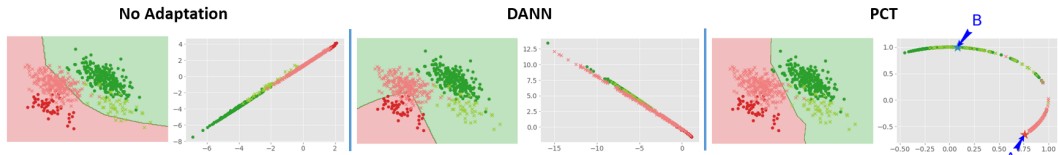

Figure 2: Visualization of different methods on a synthetic dataset, where darker points marked with "·" and lighter points marked with "×" denote the source and target samples, respectively, and the red and green colors denote two different classes. For each method, the left plot shows the data space whereas the right plot exhibits the output of the feature encoder in the latent space. Letters A and B correspond to the two class prototypes in the latent space. When there is clear class imbalance, DANN, a representative algorithm whose strategy is to match the marginal feature distributions between the source and target, fails to adapt to the target domain.

[27], and estimating the target class proportions. Our method does not introduce additional model parameters for aligning domain features, and the model can be learned in an end-to-end manner. We provide a motivating example of the application of our method on a synthetic dataset in Figure 2.

In domain adaptation, we are given a labeled dataset from the source domain, $\{(\boldsymbol{x}_i^s, y_i^s)\}_{i=1}^{n_s} \sim \mathcal{D}_s$, and an unlabeled dataset from the target domain, $\{\boldsymbol{x}_j^t\}_{j=1}^{n_t} \sim \mathcal{D}_t^{\boldsymbol{x}}$. We focus on the closed-category domain adaptation and assume that the source and target domains share the same label space, $i.e.$, $y_i^s, y_j^t \in \{1, 2, \ldots, K\}$, where $K$ denotes the number of classes. The goal of domain adaptation is to learn a model with low risk on the target samples. The model typically consists of a feature encoder, $F_{\boldsymbol{\theta}} : \mathcal{X} \to \mathbb{R}^{d_f}$, parameterized by $\boldsymbol{\theta}$, and a linear classification layer $C_{\boldsymbol{\mu}} : \mathbb{R}^{d_f} \to \mathbb{R}^K$, parameterized by $\boldsymbol{\mu}$. In prior works [6, 7, 26], the feature encoder is a pre-trained neural network, and the classifier is a randomly initialized linear layer. To simplify the following notation, we denote $\boldsymbol{f}_i^s = F_{\boldsymbol{\theta}}(\boldsymbol{x}_i^s)$ and $\boldsymbol{f}_j^t = F_{\boldsymbol{\theta}}(\boldsymbol{x}_j^t)$ as the feature representations of the source data $\boldsymbol{x}_i^s$ and target data $\boldsymbol{x}_j^t$, respectively.

## 2.1 Learning class prototypes

Most existing works [6, 7, 26] focus on aligning the source and target features in a latent space. By contrast, we propose to characterize the features of each class with a class prototype and align the target features with these class prototypes instead of the source features. This approach has several advantages. First, the feature alignment between two domains would be more robust to outliers in the source domain as we avoid using the source samples directly. Second, we do not need to worry about the missing classes in the sampled mini-batch in the source domain like we do when we align the features of source and target samples. Prototypes ensure that every class is represented for each training update. Last but not least, using the inferred prototypes instead of source features allows adapting to the target domain even without accessing the source data during the adaptation stage, which is an appealing trait when preserving the source data privacy is a concern (see Table 6).

Previous methods [28–31, 12, 32] construct each class prototype as the average latent feature for that class extracted by the feature encoder, which is computationally expensive due to the forward passing of a large number of training samples. We propose to construct class prototypes in the same latent space but with learnable parameters: $[\boldsymbol{\mu}_1, \boldsymbol{\mu}_2, \ldots, \boldsymbol{\mu}_K] \in \mathbb{R}^{d_f \times K}$, where the dimension of each prototype, $d_f$, is the same as the hidden dimension after the feature encoder $F_{\boldsymbol{\theta}}$. This strategy has been successfully applied by Saito et al. [33] in a semi-supervised learning setting. We learn each class prototype in a way that encourages the prototype to be close to the source samples associated with that class in the feature space. In particular, given the prototypes and source samples $\boldsymbol{x}_i^s$ with features $\boldsymbol{f}_i^s$ and labels $y_i^s$, we use the cross-entropy loss to learn the prototypes:

$$\mathcal{L}_{\text{cls}} = \mathbb{E}_{(\boldsymbol{x}_i^s, y_i^s) \sim \mathcal{D}_s} \left[ \sum_{k=1}^K - \log p_{ik}^s \mathbf{1}_{\{y_i^s = k\}} \right], \qquad p_{ik}^s := \frac{\exp(\boldsymbol{\mu}_k^T \boldsymbol{f}_i^s + b_k)}{\sum_{k'=1}^K \exp(\boldsymbol{\mu}_{k'}^T \boldsymbol{f}_i^s + b_{k'})}, \qquad (1)$$

where $b_k$ is a bias term and $p_{ik}^s$ is the predictive probability for $\boldsymbol{x}_i^s$ to be classified to class $k$. We note that this way of learning the class prototypes is closely connected to learning the standard linear classification layer $C_{\boldsymbol{\mu}}$ on source-only data with the cross-entropy loss. The neural network weights in the classification layer can be interpreted as the class prototypes. Therefore, compared with source-only approaches, constructing prototypes in this way introduce no additional parameters. As we show in Figure 4a, our method requires much fewer parameters than other domain-adaptation methods. Moreover, it requires much less computation than other prototype-based methods by avoiding the need to average the latent features for each class.

## 2.2 Bi-directional prototype-oriented conditional transport

In this section, we will discuss how we encourage the feature encoder to align the target data with class prototypes. Our approach is motivated by the cluster assumption, which has been widely adopted in both semi-supervised learning [20, 34, 35] and domain-adaptation literature [36, 37, 31, 38]. The cluster assumption states that the input data distribution consists of separated clusters and that instances belonging to the same cluster tend to have the same class labels. This means that the decision boundaries should not cross data high-density regions. To achieve this goal, we minimize the expected pairwise cost between the target features and prototypes with respect to two differently constructed joint distributions. By minimizing the expected costs under these two different joint distributions, the target feature will be close to the prototypes, far from the decision boundaries.

### 2.2.1 Moving from target domain to class prototypes

To define the expected cost of moving from the target domain to the class prototypes, we first use the chain rule to factorize the joint distribution of the class prototype $\boldsymbol{\mu}_k$ and target feature $\boldsymbol{f}_j^t$ as $p(\boldsymbol{f}_j^t)\pi_{\boldsymbol{\theta}}(\boldsymbol{\mu}_k \mid \boldsymbol{f}_j^t)$, where drawing from the target feature distribution $p(\boldsymbol{f}_j^t)$ can be realized by selecting a random target sample $\boldsymbol{x}_j^t \sim \mathcal{D}_t^x$ to obtain $\boldsymbol{f}_j^t = F_{\boldsymbol{\theta}}(\boldsymbol{x}_j^t)$. The conditional distribution, representing the probability of moving from target feature $\boldsymbol{f}_j^t$ to class prototype $\boldsymbol{\mu}_k$, is defined as

$$\pi_{\boldsymbol{\theta}}(\boldsymbol{\mu}_k \mid \boldsymbol{f}_j^t) = \frac{p(\boldsymbol{\mu}_k)\exp(\boldsymbol{\mu}_k^T \boldsymbol{f}_j^t)}{\sum_{k'=1}^K p(\boldsymbol{\mu}_{k'})\exp(\boldsymbol{\mu}_{k'}^T \boldsymbol{f}_j^t)}, \quad k \in \{1, \ldots, K\}, \tag{2}$$

where, through the lens of Bayes' rule, $p(\boldsymbol{\mu}_k)$ is the discrete prior distribution over the $K$ classes for the target domain, and $\exp(\boldsymbol{\mu}_k^T \boldsymbol{f}_j^t)$ plays the role of an unnormalized likelihood term, measuring the similarity between class prototypes and target features. Intuitively, the target features are more likely to be moved to the prototypes which correspond to dominant classes in the target domain or which are closer to the target features (or both). Note that in practice we often do not have access to the target class distribution $p(\boldsymbol{\mu}_k)$. We can use a uniform prior distribution for $p(\boldsymbol{\mu}_k)$. However, this could be sub-optimal, especially when classes are seriously imbalanced in the target domain. To address this issue, we propose a way to estimate $\{p(\boldsymbol{\mu}_k)\}_{k=1}^K$ in Section 2.3.

We now define the expected cost of moving the target features to class prototypes as:

$$\mathcal{L}_{t\to\mu} = \mathbb{E}_{\boldsymbol{x}_j^t \sim \mathcal{D}_t^x} \mathbb{E}_{\boldsymbol{\mu}_k \sim \pi_{\boldsymbol{\theta}}(\boldsymbol{\mu}_k \mid \boldsymbol{f}_j^t)} \left[c(\boldsymbol{\mu}_k, \boldsymbol{f}_j^t)\right] = \mathbb{E}_{\boldsymbol{x}_j^t \sim \mathcal{D}_t^x} \left[\sum_{k=1}^K c(\boldsymbol{\mu}_k, \boldsymbol{f}_j^t)\frac{p(\boldsymbol{\mu}_k)\exp(\boldsymbol{\mu}_k^T \boldsymbol{f}_j^t)}{\sum_{k'=1}^K p(\boldsymbol{\mu}_{k'})\exp(\boldsymbol{\mu}_{k'}^T \boldsymbol{f}_j^t)}\right], \tag{3}$$

where $c(\cdot, \cdot)$, a point-to-point moving cost, is defined with the cosine dissimilarity as

$$c(\boldsymbol{\mu}_k, \boldsymbol{f}_j^t) = 1 - \frac{\boldsymbol{\mu}_k^T \boldsymbol{f}_j^t}{||\boldsymbol{\mu}_k||_2 ||\boldsymbol{f}_j^t||_2}. \tag{4}$$

We also consider other point-to-point costs and present the results in Section 4.2. With Eq. (3), it is straightforward to obtain an unbiased estimation of $\mathcal{L}_{t\to\mu}$ with a mini-batch from target domain $\mathcal{D}_t^x$.

In this target-to-prototype direction, we are assigning each target sample to the prototypes according to their similarities and the class distribution. Intuitively, minimizing this expected moving cost encourages each target feature to get closer to neighboring class prototypes, reducing the violation of the cluster assumption. If we think of each prototype as the mode of the distribution of source features for a class, this loss encourages a mode-seeking behavior [27]. Still, this loss alone might lead to sub-optimal alignment. The feature encoder can map most of the target data to only a few prototypes. We connect this loss to entropy minimization to elucidate this point.

**Connection with entropy minimization.** The expected cost of moving from the target domain to prototypes can be viewed as a generalization of entropy minimization [20], an effective regularization in many prior domain-adaptation works [39, 33, 40, 17]. If the point-to-point moving cost is defined as $c(\boldsymbol{\mu}_k, \boldsymbol{f}_j^t) = -\log p_{jk}^t = -\log \frac{\exp(\boldsymbol{\mu}_k^T \boldsymbol{f}_j^t)}{\sum_{k'=1}^K \exp(\boldsymbol{\mu}_{k'}^T \boldsymbol{f}_j^t)}$ and the conditional probability is $\pi_{\boldsymbol{\theta}}(\boldsymbol{\mu}_k \mid \boldsymbol{f}_j^t) = p_{jk}^t = \frac{\exp(\boldsymbol{\mu}_k^T \boldsymbol{f}_j^t)}{\sum_{k'=1}^K \exp(\boldsymbol{\mu}_{k'}^T \boldsymbol{f}_j^t)}$ (with a uniform prior), then the expected moving cost becomes: $\mathcal{L}_{t\to\mu} = -\mathbb{E}_{\boldsymbol{x}_j^t \sim \mathcal{D}_t^x} \left[\sum_{k=1}^K p_{jk}^t \log p_{jk}^t\right]$, which is equivalent to minimizing the entropy on the target samples. Entropy minimization alone also has a mode-seeking behavior and has the same tendency to drop some modes (class prototypes). In other words, one trivial solution is to assign the same one-hot encoding to all the target samples [41, 42].

### 2.2.2 Moving from class prototypes to target domain

To ensure that each prototype has some target features located close by and avoid dropping class prototypes, we propose to add a cost of the opposite direction [27], $i.e.$, moving from the prototypes to target features. Given a mini-batch, $\{\boldsymbol{x}_j^t\}_{j=1}^M$, of target samples of size $M$, denoting $\hat{p}(\boldsymbol{f}^t) = \sum_{j=1}^M \frac{1}{M}\delta_{\boldsymbol{f}_j^t}$ as the empirical distribution of the target features in this mini-batch, the probabilities of moving from a prototype $\boldsymbol{\mu}_k$ to the $M$ target features is defined as a conditional distribution:

$$\pi_{\boldsymbol{\theta}}(\boldsymbol{f}_j^t \mid \boldsymbol{\mu}_k) = \frac{\hat{p}(\boldsymbol{f}_j^t)\exp(\boldsymbol{\mu}_k^T \boldsymbol{f}_j^t)}{\sum_{j'=1}^M \hat{p}(\boldsymbol{f}_{j'}^t)\exp(\boldsymbol{\mu}_k^T \boldsymbol{f}_{j'}^t)} = \frac{\exp(\boldsymbol{\mu}_k^T \boldsymbol{f}_j^t)}{\sum_{j'=1}^M \exp(\boldsymbol{\mu}_k^T \boldsymbol{f}_{j'}^t)}, \quad \boldsymbol{f}_j^t \in \{\boldsymbol{f}_1^t, \ldots, \boldsymbol{f}_M^t\}. \tag{5}$$

As opposed to the probabilities of moving a target feature to different class prototypes $\pi_{\boldsymbol{\theta}}(\boldsymbol{\mu}_k \mid \boldsymbol{f}_j^t)$, $\pi_{\boldsymbol{\theta}}(\boldsymbol{f}_j^t \mid \boldsymbol{\mu}_k)$ normalizes the probabilities across the $M$ target samples for each prototype, which ensures that each prototype will be assigned to some target features. Then, the expected cost of moving along this prototype-to-target direction is defined as:

$$\mathcal{L}_{\mu \to t} = \mathbb{E}_{\{\boldsymbol{x}_j^t\}_{j=1}^M \sim \mathcal{D}_t^{\boldsymbol{x}}} \mathbb{E}_{\boldsymbol{\mu}_k \sim p(\boldsymbol{\mu}_k)} \mathbb{E}_{\boldsymbol{f}_j^t \sim \pi_{\boldsymbol{\theta}}(\boldsymbol{f}_j^t \mid \boldsymbol{\mu}_k)} \left[ c(\boldsymbol{\mu}_k, \boldsymbol{f}_j^t) \right]$$

$$= \mathbb{E}_{\{\boldsymbol{x}_j^t\}_{j=1}^M \sim \mathcal{D}_t^{\boldsymbol{x}}} \left[ \sum_{k=1}^K p(\boldsymbol{\mu}_k) \sum_{j=1}^M c(\boldsymbol{\mu}_k, \boldsymbol{f}_j^t) \frac{\exp(\boldsymbol{\mu}_k^T \boldsymbol{f}_j^t)}{\sum_{j'=1}^M \exp(\boldsymbol{\mu}_k^T \boldsymbol{f}_{j'}^t)} \right], \tag{6}$$

which can be estimated by drawing a mini-batch of $M$ target samples.

Finally, combining the classification loss in Eq. (1), target-to-prototype moving cost in Eq. (3), and prototype-to-target moving cost in Eq. (6), our loss is expressed as

$$\mathcal{L}_{\text{cls}} + \mathcal{L}_{t \to \mu} + \mathcal{L}_{\mu \to t}. \tag{7}$$

Note that we treat $\boldsymbol{\mu}$ as fixed in both $\mathcal{L}_{t \to \mu}$ and $\mathcal{L}_{\mu \to t}$. This strategy allows us to apply our method in the source-data-private setting where we only have access to the source model. We also find empirically that this leads to more stable training.

### 2.3 Learning class proportions in the target domain

We propose to infer the class proportions $\{p(\boldsymbol{\mu}_k)\}_{k=1}^K$ in the target domain by maximizing the log-likelihood of the unlabeled target data while fixing the class prototypes $\boldsymbol{\mu}$. Directly optimizing the marginal likelihood is intractable, so we use the EM algorithm [43–45] to derive the following iterative updates (see the derivation in Appendix B). We first initialize with a uniform prior: $p(\boldsymbol{\mu}_k)^0 = \frac{1}{K}$, and obtain new estimates at each update step $l$ (starting from 0):

$$p(\boldsymbol{\mu}_k)^{l+1} = \frac{1}{M}\sum_{j=1}^M \pi_{\boldsymbol{\theta}}^l(\boldsymbol{\mu}_k \mid \boldsymbol{f}_j^t), \quad \text{where} \quad \pi_{\boldsymbol{\theta}}^l(\boldsymbol{\mu}_k \mid \boldsymbol{f}_j^t) = \frac{p(\boldsymbol{\mu}_k)^l \exp(\boldsymbol{\mu}_k^T \boldsymbol{f}_j^t)}{\sum_{k'=1}^K p(\boldsymbol{\mu}_{k'})^l \exp(\boldsymbol{\mu}_{k'}^T \boldsymbol{f}_j^t)}. \tag{8}$$

Intuitively, the average predicted probabilities over the target examples for each class are used to estimate the target proportions, with $p(\boldsymbol{\mu}_k)^{l+1}$ shown above providing an estimate based on a single mini-batch of $M$ target samples. To estimate it based on the full dataset, we iteratively update it with $p(\boldsymbol{\mu}_k)^{l+1} \leftarrow (1 - \beta^l)p(\boldsymbol{\mu}_k)^l + \beta^l p(\boldsymbol{\mu}_k)^{l+1}$, where we follow the decaying schedule of the learning rate of the other parameters to set $\beta^l = \beta_0(1 + \gamma l)^{-\alpha}$, in which $\gamma = 0.0002$, $\alpha = 0.75$. The inintial value $\beta_0$ is a hyper-parameter that can be set as either 0 to indicate a uniform prior, or a small value, such as 0.001, to allow the class proportions to be inferred from the data.

## 3 Related work

**Feature distribution alignment.** Most works on domain adaptation with deep learning focus on feature alignment to align either the marginal distributions [46, 6, 8, 47, 48] or the joint distributions [7, 26] of the deep features of neural networks. Early works use adversarial-based objectives, which are equivalent to minimizing the Jensen–Shannon (JS) divergence [49]. When the source and target domains have non-overlapping supports, the JS divergence fails to supply useful gradients [50, 51]. To remedy this issue, researchers propose using the Wasserstein distance, which arises from the optimal transport problem [52]. Courty et al. [9] develop Joint Distribution Optimal Transport (JDOT), defining the transport cost to be the joint cost in the data and prediction spaces. Several works [10, 12] extend this framework for deep neural networks. However, solving for the optimal couplings without

any relaxation is a linear programming problem, which has a complexity of $\mathcal{O}(M^3 \log M)$ [53]. Computing the transport probabilities in our algorithm has a complexity of $\mathcal{O}(d_f M K)$, which is the same complexity as computing the predictive probabilities that need to be computed by most algorithms. In this category, all the works focus on aligning the target features with source features, whereas we align the target features with prototypes.

**Prototype-based alignment.** We make two distinctions to existing works in this area. First, prior domain-adaptation works for classification [29–31] and segmentation [54, 55] utilize prototypes for pseudo-label assignments. By contrast, we use prototypes to behave as representative samples of the source features to define the loss. Second, all of these works use some form of average latent features to construct class prototypes, which is computationally expensive. We instead adopt a parametric approach to learn the prototypes, avoiding that costly computation.

**Learning under shifted class proportions.** Although a shift in class proportions between two domains is a common problem in many applications, it is still largely under-explored. Over the years, researchers have viewed the question through different lenses: applying kernel distribution embedding [56, 57, 21], using an EM update [43, 58], placing a meta-prior over the target proportions [59], and casting the problem under causal and anti-causal learning [21–25]. Recently, Tachet des Combes et al. [18] propose aligning the target feature distribution with a re-weighted version of the source feature distribution. While this method achieves consistent improvements over feature-alignment algorithms, it still relies on learning domain-invariant representations. As we have discussed, this approach suffers from the problems of sampling variability and class-mismatching in a mini-batch, whereas the proposed method uses class prototypes and proportion estimation to avoid these issues.

**Source-private adaptation.** Finally, the proposed method can also be applied to a source-private setting where without seeing the raw source data, we only have access to the source model and target data while adapting to the target domain [31, 47, 60–63]. Liang et al. [31] introduce a clustering-based approach to generate pseudo-labels for the target data. That approach requires constructing class centers using a weighted average of the latent features. Different from that work, our prototypes behave as class centers and are more amenable to mini-batch stochastic gradient based training.

# 4 Experiments

In this section, we evaluate our method under four practical settings: single-source, multi-source, class-imbalance, and source-private domain adaptation. We present the setup, the results, and the analysis of the results in the upcoming sections.

**Datasets.** We use the following three datasets of varying sizes in our experiment: **1)** *Office-31* [3] consists of $4652$ images coming from three domains: Amazon (A), Webcam (W), and DSLR (D). The total number of categories is $31$. **2)** *Office-Home* [64], a more challenging dataset than Office-31, consists of 15,500 images over 65 classes and four domains: Artistic images (Ar), Clip art (Cl), Product images (Pr), and Real-world (Rw). **3)** *DomainNet* [65] is a large-scale dataset for domain adaptation. It consists of about 569,010 images with 345 categories from six domains: Clipart, Infograph, Painting, Quickdraw, Real, and Sketch. We further perform experiments on Cross-Digits, ImageClef, Office-Caltech, and VisDA datasets and provide the details in Appendix E.2.

**Implementation details.** We implement our method on top of the open-source transfer learning library (MIT license) [66], adopting the default neural network architectures for both the feature encoder and linear classifier. For the feature encoder network, we utilize a pre-trained ResNet-50 in all experiments except for multi-source domain adaptation, where we use a pre-trained ResNet-101. We fine-tune the feature encoder and train the linear layers from random initialization. The linear layers have the learning rate of $0.01$, 10 times that of the feature encoder. The learning rate follows the following schedule as $\eta_{\text{iter}} = \eta_0 (1 + \gamma \text{iter})^{-\alpha}$, where $\eta_0$ is the initial learning rate. We set $\eta_0$ to $0.01$, $\gamma$ to $0.0002$, and $\alpha$ to $0.75$. We utilize a mini-batch SGD with a momentum of $0.9$. We set the batch size for the source data as $N = 32$ and that for the target data as $M = 96$. We use all the labeled source samples and unlabeled target samples [6, 7, 26]. We set $\beta_0$ to $0$ (a uniform prior) in all settings except for the sub-sampled target datasets. We perform a sensitivity analysis (see Appendix E) and set $\beta_0$ empirically to $0.001$ for the sub-sampled target version of Office-31 and $0.0001$ for that of Office-Home. We report the average accuracy from three independent runs. All experiments are conducted using a single Nvidia Tesla V100 GPU except for the DomainNet experiment, where we use four V100 GPUs. More implementation details can be found in Appendix D.

### 4.1 Main results

**Single-source setting.** We perform single-source domain adaptation on the Office-31 and Office-Home datasets. In each experiment, one domain serves as the source domain and another as the target domain. We consider all permutations, leading to 6 tasks for the Office-31 dataset and 12 tasks for the Office-Home dataset. We compare our algorithm with state-of-the-art algorithms for domain adaptations from three different categories: adversarial-based, divergence-based, and optimal transport-based. We report the results on Office-31 in Table 1. PCT significantly outperforms the baselines, especially on the more difficult transfer tasks (D→A and W → A). Although MDD [67], the best baseline domain-adaptation method, uses a bigger classifier, PCT still has $1.1\%$ higher average accuracy. In Figure 4a, we visualize the number of parameters versus the average accuracy on the Office-31 dataset. While PCT uses fewer parameters than most methods, it still achieves the highest average accuracy. We report the average accuracies on the Office-Home dataset in Table 2. PCT outperforms baseline methods on 10 of the 12 transfer tasks, yielding $3.7\%$ improvement on the average accuracy over MDD. The results in this setting demonstrate that aligning the target features with prototypes is more effective than directly aligning them with the source features.

Table 1: Accuracy (%) on Office-31 for unsupervised domain adaptation (ResNet-50).

| Category | Method | A → W | D → W | W → D | A → D | D → A | W → A | Avg |
|---|---|---|---|---|---|---|---|---|
| | ResNet-50 [68] | $68.4 \pm 0.2$ | $96.7 \pm 0.1$ | $99.3 \pm 0.1$ | $68.9 \pm 0.2$ | $62.5 \pm 0.3$ | $60.7 \pm 0.3$ | 76.1 |
| Adversarial | DANN [6] | $82.0 \pm 0.4$ | $96.9 \pm 0.2$ | $99.1 \pm 0.1$ | $79.7 \pm 0.4$ | $68.2 \pm 0.4$ | $67.4 \pm 0.5$ | 82.2 |
| | ADDA [47] | $86.2 \pm 0.5$ | $96.2 \pm 0.3$ | $98.4 \pm 0.3$ | $77.8 \pm 0.3$ | $69.5 \pm 0.4$ | $68.9 \pm 0.5$ | 82.9 |
| | CDAN [26] | $94.1 \pm 0.1$ | $98.6 \pm 0.1$ | $\mathbf{100.0} \pm 0.0$ | $92.9 \pm 0.2$ | $71.0 \pm 0.3$ | $69.3 \pm 0.3$ | 87.7 |
| | MDD [67] | $94.5 \pm 0.3$ | $98.4 \pm 0.1$ | $\mathbf{100.0} \pm 0.0$ | $93.5 \pm 0.2$ | $74.6 \pm 0.3$ | $72.2 \pm 0.1$ | 88.9 |
| Divergence | JAN [7] | $85.4 \pm 0.3$ | $97.4 \pm 0.2$ | $99.8 \pm 0.2$ | $84.7 \pm 0.3$ | $68.6 \pm 0.3$ | $70.0 \pm 0.4$ | 84.3 |
| | TPN [29] | $91.2 \pm 0.3$ | $97.7 \pm 0.2$ | $99.5 \pm 0.1$ | $89.9 \pm 0.2$ | $70.5 \pm 0.2$ | $73.5 \pm 0.1$ | 87.1 |
| OT | DeepJDOT [10] | $88.9 \pm 0.3$ | $98.5 \pm 0.1$ | $99.6 \pm 0.2$ | $88.2 \pm 0.1$ | $72.1 \pm 0.4$ | $70.1 \pm 0.4$ | 86.2 |
| | ETD [69] | 92.1 | $\mathbf{100.0}$ | $\mathbf{100.0}$ | 88.0 | 71.0 | 67.8 | 86.2 |
| | PCT (Ours) | $\mathbf{94.6} \pm 0.5$ | $98.7 \pm 0.4$ | $99.9 \pm 0.1$ | $\mathbf{93.8} \pm 1.8$ | $\mathbf{77.2} \pm 0.5$ | $\mathbf{76.0} \pm 0.9$ | $\mathbf{90.0}$ |

Table 2: Accuracy (%) on Office-Home for unsupervised domain adaptation (ResNet-50).

| Method | Ar → Cl | Ar → Pr | Ar → Rw | Cl → Ar | Cl → Pr | Cl → Rw | Pr → Ar | Pr → Cl | Pr → Rw | Rw → Ar | Rw → Cl | Rw → Pr | Avg |
|---|---|---|---|---|---|---|---|---|---|---|---|---|---|
| ResNet-50 [68] | 34.9 | 50.0 | 58.0 | 37.4 | 41.9 | 46.2 | 38.5 | 31.2 | 60.4 | 53.9 | 41.2 | 59.9 | 46.1 |
| DANN [6] | 45.6 | 59.3 | 70.1 | 47.0 | 58.5 | 60.9 | 46.1 | 43.7 | 68.5 | 63.2 | 51.8 | 76.8 | 57.6 |
| CDAN [26] | 50.7 | 70.6 | 76.0 | 57.6 | 70.0 | 70.0 | 57.4 | 50.9 | 77.3 | 70.9 | 56.7 | 81.6 | 65.8 |
| MDD [67] | 54.9 | 73.7 | 77.8 | 60.0 | 71.4 | 71.8 | 61.2 | 53.6 | 78.1 | 72.5 | $\mathbf{60.2}$ | 82.3 | 68.1 |
| JAN [7] | 45.9 | 61.2 | 68.9 | 50.4 | 59.7 | 61.0 | 45.8 | 43.4 | 70.3 | 63.9 | 52.4 | 76.8 | 58.3 |
| TPN [29] | 51.2 | 71.2 | 76.0 | 65.1 | 72.9 | 72.8 | 55.4 | 48.9 | 76.5 | 70.9 | 53.4 | 80.4 | 66.2 |
| DeepJDOT [10] | 48.2 | 69.2 | 74.5 | 58.5 | 69.1 | 71.1 | 56.3 | 46.0 | 75.5 | 68.0 | 52.7 | 80.9 | 64.3 |
| ETD [69] | 51.3 | 71.9 | $\mathbf{85.7}$ | 57.6 | 69.2 | 73.7 | 57.8 | 51.2 | 79.3 | 70.2 | 57.5 | 82.1 | 67.3 |
| PCT (Ours) | $\mathbf{57.1}$ ±0.3 | $\mathbf{78.3}$ ±1.2 | 81.4 ±0.4 | $\mathbf{67.6}$ ±0.3 | $\mathbf{77.0}$ ±1.2 | $\mathbf{76.5}$ ±0.5 | $\mathbf{68.0}$ ±0.5 | $\mathbf{55.0}$ ±0.6 | $\mathbf{81.3}$ ±0.2 | $\mathbf{74.7}$ ±0.5 | 60.0 ±0.5 | $\mathbf{85.3}$ ±0.3 | $\mathbf{71.8}$ |

**Multi-source setting.** In this setting, we evaluate our method using Office-Home and DomainNet datasets [65]. For each task, we select one domain as the target domain and use the remaining five domains as the source. We use the same data splitting scheme as the original paper [65]. We compare against source-combined and multi-source algorithms introduced in Venkat et al. [70] for Office-Home and in Peng et al. [65] for DomainNet. Multi-source algorithms use domain labels and a classifier for each source domain, whereas source-combined algorithms combine all the source domains into a single source domain and perform single-source adaptation. We adopt a single classifier and do not use domain labels. Thus, PCT falls under the source-combined category. We report the results in Tables 3 and 4. While PCT is not designed specifically for multi-source domain adaptation, our method still outperforms those multi-source algorithms in both datasets. Since there are more variations of the data in the source domain in this setting, the increase in performance gain verifies our intuition that prototypes help mitigate the problem of sampling variability.

Table 3: Accuracy (%) on Office-Home for ResNet50-based MSDA methods.

| Category | Models | $\mathcal{R} \to \text{Ar}$ | $\mathcal{R} \to \text{Cl}$ | $\mathcal{R} \to \text{Pr}$ | $\mathcal{R} \to \text{Rw}$ | Avg |
|---|---|---|---|---|---|---|
| Source-combined | DAN [8] | 68.5 | 59.4 | 79.0 | 82.5 | 72.4 |
| | D-CORAL [71] | 68.1 | 58.6 | 79.5 | 82.7 | 72.2 |
| | RevGrad [6] | 68.4 | 59.1 | 79.5 | 82.7 | 72.4 |
| Multi-source | MFSAN [72] | 72.1 | 62.0 | 80.3 | 81.8 | 74.1 |
| | SImpAl [70] | 70.8 | 56.3 | 80.2 | 81.5 | 72.2 |
| | PCT (Ours) | $\mathbf{76.3} \pm 0.5$ | $\mathbf{64.1} \pm 0.4$ | $\mathbf{84.9} \pm 0.8$ | $\mathbf{84.3} \pm 0.5$ | $\mathbf{77.4}$ |

Table 4: Accuracy (%) on DomainNet for ResNet101-based MSDA methods.

| Category | Models | $\mathcal{R} \rightarrow$ Clipart | $\mathcal{R} \rightarrow$ Infograph | $\mathcal{R} \rightarrow$ Painting | $\mathcal{R} \rightarrow$ Quickdraw | $\mathcal{R} \rightarrow$ Real | $\mathcal{R} \rightarrow$ Sketch | Avg |
|---|---|---|---|---|---|---|---|---|
| Multi-source | DCTN [73] | $48.6 \pm 0.7$ | $23.5 \pm 0.6$ | $48.8 \pm 0.6$ | $7.2 \pm 0.4$ | $53.5 \pm 0.6$ | $47.3 \pm 0.5$ | $38.2 \pm 0.6$ |
| | M$^3$ SDA [65] | $57.2 \pm 1.0$ | $24.2 \pm 1.2$ | $51.6 \pm 0.4$ | $5.2 \pm 0.5$ | $61.6 \pm 0.9$ | $49.6 \pm 0.6$ | $41.5 \pm 0.7$ |
| | M$^3$ SDA-$\beta$ [65] | $58.6 \pm 0.5$ | $26.0 \pm 0.9$ | $52.3 \pm 0.6$ | $6.3 \pm 0.6$ | $62.7 \pm 0.5$ | $49.5 \pm 0.8$ | $42.6 \pm 0.6$ |
| | ML-MSDA [74] | $61.4 \pm 0.8$ | $\mathbf{26.2} \pm 0.4$ | $51.9 \pm 0.2$ | $\mathbf{19.1} \pm 0.3$ | $57.0 \pm 1.0$ | $50.3 \pm 0.7$ | $44.3 \pm 0.6$ |
| Source-combined | ResNet-101 [24] | $47.6 \pm 0.5$ | $13.0 \pm 0.4$ | $38.1 \pm 0.5$ | $13.3 \pm 0.4$ | $51.9 \pm 0.9$ | $33.7 \pm 0.5$ | $32.9 \pm 0.5$ |
| | DAN [8] | $45.4 \pm 0.5$ | $12.8 \pm 0.9$ | $36.2 \pm 0.6$ | $15.3 \pm 0.4$ | $48.6 \pm 0.7$ | $34.0 \pm 0.5$ | $32.1 \pm 0.6$ |
| | RTN [75] | $44.2 \pm 0.6$ | $12.6 \pm 0.7$ | $35.3 \pm 0.6$ | $14.6 \pm 0.8$ | $48.4 \pm 0.7$ | $31.7 \pm 0.7$ | $31.1 \pm 0.7$ |
| | JAN [7] | $40.9 \pm 0.4$ | $11.1 \pm 0.6$ | $35.4 \pm 0.5$ | $12.1 \pm 0.7$ | $45.8 \pm 0.6$ | $32.3 \pm 0.6$ | $29.6 \pm 0.6$ |
| | DANN [6] | $45.5 \pm 0.6$ | $13.1 \pm 0.7$ | $37.0 \pm 0.7$ | $13.2 \pm 0.8$ | $48.9 \pm 0.7$ | $31.8 \pm 0.6$ | $32.6 \pm 0.7$ |
| | ADDA [47] | $47.5 \pm 0.8$ | $11.4 \pm 0.7$ | $36.7 \pm 0.5$ | $14.7 \pm 0.5$ | $49.1 \pm 0.8$ | $33.5 \pm 0.5$ | $32.2 \pm 0.6$ |
| | SE [38] | $24.7 \pm 0.3$ | $3.9 \pm 0.5$ | $12.7 \pm 0.4$ | $7.1 \pm 0.5$ | $22.8 \pm 0.5$ | $9.1 \pm 0.5$ | $16.1 \pm 0.4$ |
| | MCD [76] | $54.3 \pm 0.6$ | $22.1 \pm 0.7$ | $45.7 \pm 0.6$ | $7.6 \pm 0.5$ | $58.4 \pm 0.7$ | $43.5 \pm 0.6$ | $38.5 \pm 0.6$ |
| | PCT (Ours) | $\mathbf{67.2} \pm 0.5$ | $26.1 \pm 0.2$ | $\mathbf{55.0} \pm 0.2$ | $16.2 \pm 0.2$ | $\mathbf{67.1} \pm 0.2$ | $\mathbf{53.7} \pm 0.6$ | $\mathbf{47.6} \pm 0.1$ |

**Sub-sampled setting.** In many cases, the label proportions could significantly change from one dataset to another, resulting in a decrease in a model's performance. To test our algorithm under this setting, we follow the experimental protocol in Tachet des Combes et al. [18]. We keep only thirty percent of the first $\lfloor K/2 \rfloor$ classes to simulate class imbalance. We directly take their results for the sub-sampled source data and perform additional experiments using the same sub-sampling scheme on the target data. The baselines in this setting are standard domain adaptation methods (DAN, JAN, and CDAN) and their importance weighted versions introduced in Tachet des Combes et al. [18]. We present the results in Table 5. On the sub-sampled source data, PCT with uniform prior outperforms the second-best method (IWCDAN) by $4\%$ on Office-31 and $6.6\%$ on Office-Home. Learning the prior distribution on the target domain does not improve the result, as this setting does not have a serious imbalance issue in the target domain. On the sub-sampled target data, PCT with a uniform prior already outperforms the baselines, $1.9\%$ and $5.2\%$ higher average accuracy than IWCDAN's on Office-31 and Office-Home, respectively. Using a learnable prior further improves upon using a uniform prior by $1.0\%$ on Office-31 and by $0.4\%$ on Office-Home. The improvements confirm our intuition that prototypes help with the class imbalance in both the source and target domain while estimating the target proportion further boosts the performance in the target domain sub-sampled setting. We visualize the estimated proportions on the target data in Figure 3, verifying that the proportions are inferred correctly.

Table 5: Average accuracy (%) on sub-sampled version of Office-31 and Office-Home (ResNet-50).

| Method | sub-S O-31 | sub-T O-31 | sub-S O-H | sub-T O-H |
|---|---|---|---|---|
| ResNet-50 [68] | 75.7 | 76.1 | 51.4 | 58.2 |
| DANN [8] | 76.2 | 75.9 | 51.8 | 58.3 |
| JAN [7] | 78.2 | 78.1 | 53.9 | 61.4 |
| CDAN [26] | 81.6 | 83.0 | 56.3 | 63.1 |
| IWDAN [18] | 82.6 | 79.2 | 57.6 | 58.6 |
| IWJAN [18] | 82.6 | 82.8 | 55.9 | 62.0 |
| IWCDAN [18] | 83.9 | 83.5 | 61.2 | 64.6 |
| PCT-Uniform (Ours) | $\mathbf{87.9} \pm 0.4$ | $85.4 \pm 0.3$ | $\mathbf{67.8} \pm 0.3$ | $69.8 \pm 0.2$ |
| PCT-Learnable (Ours) | $\mathbf{87.9} \pm 0.4$ | $\mathbf{86.4} \pm 0.2$ | $\mathbf{67.8} \pm 0.3$ | $\mathbf{70.2} \pm 0.2$ |

**Source-data-private setting.** In many practical applications, practitioners might not directly have access to the source data in the adaptation stage. Instead, a trained model on the source data is provided. In this setting, the goal is to adapt to the target domain while only operating on the given model. We compare our method with Source Hypothesis Transfer (SHOT) [31], a state-of-the-art method proposed specifically for this setting. SHOT contains two losses: an information maximization (IM) loss and a pseudo-labeling loss. We follow their experimental protocol by first training the model on the source data alone. During the adaptation stage, we only use the target data to perform model adaptation. We use the transport losses to update the feature encoder while fixing the prototypes. We report the results in Table 6. From the standard setting where we have access to source data in Table 1, the average accuracy drops by $1.6\%$ for Office-31 and by $0.8\%$ for Office-Home. On the Office-31 dataset, our bi-directional loss outperforms the IM loss by $1.1\%$ and the pseudo label loss by $0.8\%$. While the average accuracy for our method is $0.2\%$ lower than both of their losses combined, the $p$-value for the independent two-sample $t$-test on the accuracies of different runs is $0.32$, which is not statistically significant. On the Office-Home dataset, our approach performs $1.9\%$ and $0.5\%$ better than the pseudo label and IM losses, respectively. While their combined loss achieves $0.8\%$ accuracy higher than that of our method, the pseudo-labeling loss in SHOT requires

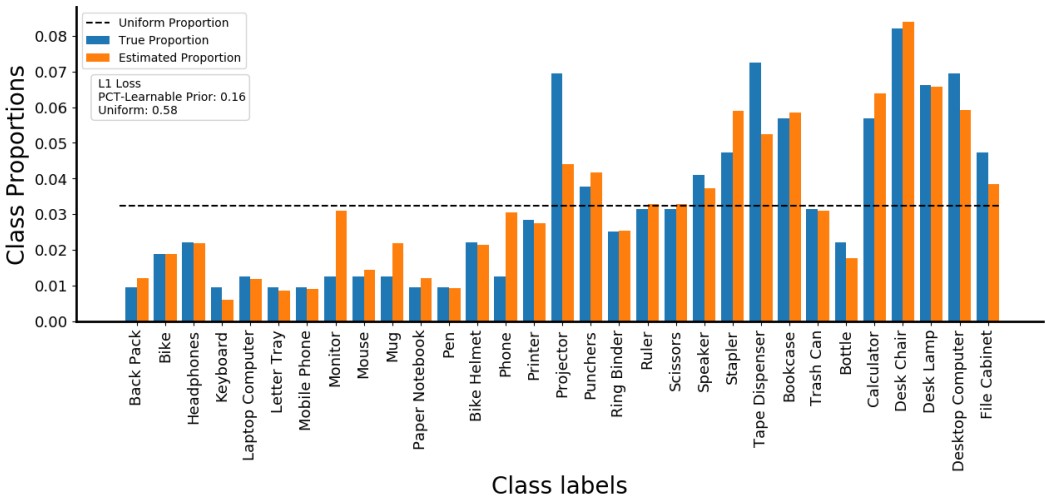

Figure 3: Visualization of the estimated target proportions versus true class proportions for the task A → sD on the Office-31 dataset. The dotted line represents a uniform proportion. It is clear that each orange bar (the learned proportions) is close to its adjacent blue bar (the true proportion). To quantify this observation, we measure the L1 loss between the true and learned proportions. The estimated proportions achieve lower L1 loss than the uniform distribution (0.16 vs 0.58), illustrating the effectiveness of our estimation strategy.

Table 6: Average Accuracy (%) on the source-private Office-31 and Office-Home (ResNet-50).

|  | Source Model Only | SHOT-Pseudo Label [31] | SHOT-IM [31] | SHOT [31] | PCT (Ours) |
|---|---|---|---|---|---|
| Office-31 | 79.3 | 87.6 ±0.5 | 87.3 ±0.5 | **88.6** ± 0.4 | 88.4 ±0.6 |
| Office-Home | 60.2 | 69.1 ±0.6 | 70.5±0.3 | **71.8** ± 0.4 | 71.0 ±0.6 |

constructing class centers, which does not scale well with large datasets. Our approach uses the classifier's weights as class prototypes to avoid this issue.

## 4.2 Analysis of results

**Ablation study.** To examine the effect of each component in our framework, we perform ablation studies and present the results in Table 7. **1)** *Alignment strategy.* Next, we present the result using optimal transport as the alignment strategy. We consider two variants of Prototype-oriented Optimal Transport (POT): exact linear program (POT) and Sinkhorn relaxation (POT-Sinkhorn). In each variant, we solve for the optimal couplings using the optimal transport formulation. After obtaining the transport probabilities, we update the feature encoder using the obtained probabilities as the weights for the transport cost. We can see that POT performs $1.7\%$ better than DeepJDOT in Table 1, showing the effectiveness of using prototypes to define the transport costs with the target features. Still, both versions of POT underperforms PCT by $2.1\%$ and $1.7\%$, respectively. **2)** *Effect of each loss in PCT.* We examine the effect of each loss on the average test accuracy on the Office-31 dataset. We remove each transport loss while keeping the cross-entropy loss. The bi-directional loss leads to the best accuracy, while the drop in accuracy is more significant if we remove $\mathcal{L}_{\mu \to t}$. This result is not surprising because, without $\mathcal{L}_{\mu \to t}$, the model can map target data to only a few prototypes, leading to a degenerate solution. **3)** *Gradient stopping.* We also show the algorithm's performance without stopping the gradient of $\boldsymbol{\mu}$ in the transport loss. PCT gains an additional $1.0\%$ in accuracy with the gradient stopping strategy. The performance gain is consistent with the finding in recent work by Chen and He [77], where the authors apply the gradient stopping strategy to avoid degenerate solutions in contrastive learning. **4)** *Cost function.* Finally, we explore the cost function inspired by the radial basis kernel. We can see that the cosine distance in PCT gives $3.1\%$ higher average accuracy. In short, the choice of the probabilistic bi-directional transport framework, gradient-stopping strategy, and point-to-point cost function all contribute to the success of the proposed PCT method.

Table 7: Average accuracy (%) of PCT on Office-31 under different variants (ResNet-50).

| POT | POT-Sinkhorn | PCT w/o ($\mathcal{L}_{t\to\mu}$) | PCT w/o ($\mathcal{L}_{\mu\to t}$) | w/o stop-grad | $c(\boldsymbol{\mu}_k, \boldsymbol{f}_j^t) = \exp(-\boldsymbol{\mu}_k^T \boldsymbol{f}_j^t)$ | PCT (Ours) |
|---|---|---|---|---|---|---|
| $87.9 \pm 0.8$ | $88.3 \pm 0.9$ | $88.6 \pm 0.2$ | $84.3 \pm 0.9$ | $89.0 \pm 0.3$ | $86.9 \pm 0.4$ | $90.0 \pm 0.5$ |

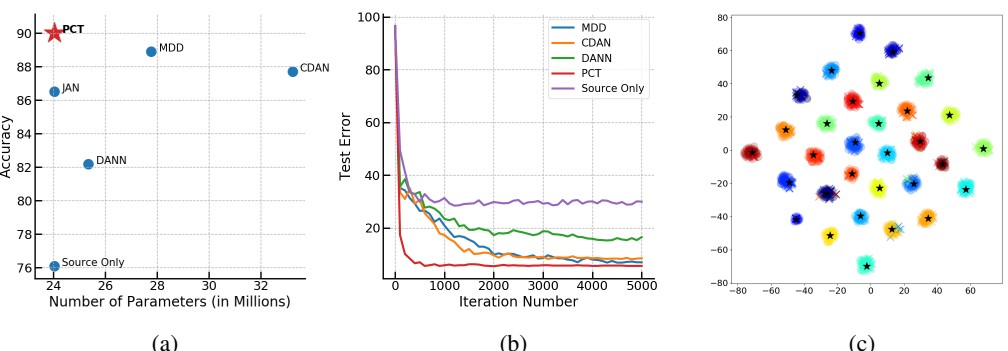

|       (a)        |        (b)        |        (c)        |

Figure 4: (a) Analysis of parameter efficiency, (b) comparison of convergence, and (c) a t-SNE visualization of the output of the feature encoder trained with PCT on the task A $\to$ W. In plot (c), prototypes ($\star$), source features ($\cdot$), and target features ($\times$) are tightly clustered together for each class.

**Convergence comparison.** We plot test accuracy versus iteration number on the task (A $\to$ W) in Figure 4b to compare the convergence rate of different algorithms. We also visualize test accuracy versus convergence time in minutes in Appendix E. In both plots, PCT quickly converges within the first one thousand iterations, and the test accuracy does not fluctuate much thereafter. This phenomenon is not surprising since we use prototypes instead of the source features to align with the target features. We expect the prototypes to behave as representative samples of the source features, making the model converge quickly and stably.

**Visualization.** We visualize in Figure 4c the t-SNE plot of the source and target features as well as the prototypes for the task A $\to$ W. Figure 4c shows that both the source (dots $\cdot$) and target (crosses $\times$) features are close to the prototypes (black stars $\star$), indicating that our algorithm is learning meaningful prototypes and successfully align the target features with the prototypes.

## 5 Conclusion

We offer a holistic framework for unsupervised domain adaptation through the lens of a probabilistic bi-directional transport between the target features and class prototypes. With extensive experiments under various application scenarios of unsupervised domain adaptation, we show that the proposed prototype-oriented alignment method works well on multiple datasets, is robust against class imbalance, and can perform domain adaptation with no direct access to the source data. Without adding additional model parameters, our memory and computation-efficient algorithm achieves competitive performance with state-of-the-art methods on several widely used benchmarks.

## Acknowledgments

We thank Camillia Smith Barnes and Georgii Riabov for helpful discussions and feedback on the paper. K. Tanwisuth, X. Fan, H. Zheng, S. Zhang, and M. Zhou acknowledge the support of Grant IIS-1812699 from the U.S. National Science Foundation, the APX 2019 project sponsored by the Office of the Vice President for Research at The University of Texas at Austin, the support of a gift fund from ByteDance Inc., and the Texas Advanced Computing Center (TACC) for providing HPC resources that have contributed to the research results reported within this paper.

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
