# A Prototype-Oriented Framework for Unsupervised Domain Adaptation: Appendix

**Korawat Tanwisuth[1], Xinjie Fan[1], Huangjie Zheng[1], Shujian Zhang[1],**

**Hao Zhang[2], Bo Chen[3], Mingyuan Zhou[1]**

[1]The University of Texas at Austin    [2] Cornell University    [3]Xidian University

## A  Broader impact

Any methods that deal with classification can suffer from dataset bias caused by domain shift between training and testing data. While domain adaptation can help mitigate the problem [78], it cannot eliminate the issue because of the combinatorial nature of too many exogenous factors [79]. Also, as with any computationally intensive endeavor, care must be taken to use sustainable energy sources. On a positive note, our method addresses many practical problems—including sampling variability, class-imbalance, and source-data privacy—by leveraing class prototypes.

## B  EM steps derivation

Since the target label of each data point, $y_j^t$, is not observed, we can view it as a latent variable. The unknown quantity we are interested in is the proportion of classes in the target data $p(\boldsymbol{\mu}_k) = p(y_j^t = k)$. To infer this quantity, we maximize the likelihood on the observed target data:

$$l(\{p(\boldsymbol{\mu}_k)\}_{k=1}^K \,|\, \boldsymbol{x}_1^t, \boldsymbol{x}_2^t, \ldots, \boldsymbol{x}_{n_t}^t) = \sum_{j=1}^{n_t} \ln[\sum_{k=1}^K p(\boldsymbol{\mu}_k) p_{\boldsymbol{\theta},\boldsymbol{\mu}}(\boldsymbol{x}_j|y_j^t = k)], \tag{9}$$

where $p_{\boldsymbol{\theta},\boldsymbol{\mu}}(\boldsymbol{x}_j|y_j^t = k) = \frac{\exp(\boldsymbol{\mu}_k^T \boldsymbol{f}_j^t)}{Z}$ and $Z$ is a normalizing constant. Note that $\boldsymbol{\mu}_k$ are learned jointly using the cross entropy loss on the source data.

Since it is difficult to directly optimize the marginal likelihood due to the sum inside the log function, we resort to the Expectation-Maximization (EM) algorithm, where we iterate between the expectation and maximization steps. We first initialize $p(\boldsymbol{\mu}_k)$ with a uniform prior , $p(\boldsymbol{\mu}_k)^0 = \frac{1}{K}, \quad \forall k = 1, 2, \ldots, K$, before performing the iterative updates. At each step $l$ (starting from 0), we conduct the E-step and M-step as follows:

E-step: Compute the posterior probability of the target data belonging to class $k$ using the old estimates. The posterior probabilities correspond to the weights of the transport cost of moving from target features to the prototypes.

$$p_{\boldsymbol{\theta},\boldsymbol{\mu}}(y_j^t = k|\boldsymbol{x}_j^t, p(\boldsymbol{\mu}_k)^l) = \pi_{\boldsymbol{\theta}}(\boldsymbol{\mu}_k \,|\, \boldsymbol{f}_j^t, p(\boldsymbol{\mu}_k)^l) = \frac{p(\boldsymbol{\mu}_k)^l \exp(\boldsymbol{\mu}_k^T \boldsymbol{f}_j^t)}{\sum_{k'=1}^K p(\boldsymbol{\mu}_{k'})^l \exp(\boldsymbol{\mu}_{k'}^T \boldsymbol{f}_j^t)}$$

M-step: The log-likelihood of the complete data is given by $\sum_{j=1}^M \ln[p(\boldsymbol{\mu}_k)^{l+1} p_{\boldsymbol{\theta},\boldsymbol{\mu}}(\boldsymbol{x}_j|y_j^t)]$. We maximize the expected complete log-likelihood where the expectation is taken with respect to the posterior distribution found in the E-step:

$$p(\boldsymbol{\mu}_k)^{l+1} = \underset{p(\boldsymbol{\mu}_k)^{l+1}}{\mathrm{argmax}}\, L \tag{10}$$

$$:= \underset{p(\boldsymbol{\mu}_k)^{l+1}}{\mathrm{argmax}} \sum_{j=1}^{n_t} \sum_{k=1}^K p_{\boldsymbol{\theta},\boldsymbol{\mu}}(y_j^t = k|\boldsymbol{x}_j^t, p(\boldsymbol{\mu}_k)^l) \ln[p(\boldsymbol{\mu}_k)^{l+1} p_{\boldsymbol{\theta},\boldsymbol{\mu}}(\boldsymbol{x}_j|y_j^t = k)] + \lambda(\sum_{k=1}^K p(\boldsymbol{\mu}_k)^{l+1} - 1) \tag{11}$$

Here, $\lambda$ is a Lagrange multiplier to enforce the constraint that $p(\boldsymbol{\mu}_k)$ should lie in a simplex.

$$\frac{\partial L}{\partial \lambda} = \sum_{k=1}^{K} p(\boldsymbol{\mu}_k)^{l+1} - 1 = 0$$

$$\frac{\partial L}{\partial p(\boldsymbol{\mu}_k)^{l+1}} = \sum_{j=1}^{n_t} p_{\boldsymbol{\theta},\boldsymbol{\mu}}(y_j^t = k | \boldsymbol{x}_j^t, p(\boldsymbol{\mu}_k)^l) \frac{1}{p(\boldsymbol{\mu}_k)^{l+1}} + \lambda = 0$$

Multiplying both sides by $p(\boldsymbol{\mu}_k)^{l+1}$ on and summing over k, we obtain the following update rule for $p(\boldsymbol{\mu}_k)^{l+1}$. And we get:

$$\lambda = -n_t,$$

$$p(\boldsymbol{\mu}_k)^{l+1} = \frac{1}{n_t} \sum_{j=1}^{n_t} p_{\boldsymbol{\theta},\boldsymbol{\mu}}(y_j^t = k | \boldsymbol{x}_j^t, p(\boldsymbol{\mu}_k)^l) = \frac{1}{n_t} \sum_{j=1}^{n_t} \pi(\boldsymbol{\mu}_k | \boldsymbol{f}_j^t, p(\boldsymbol{\mu}_k)^l).$$

In practice, we draw a mini-batch of size $M$ to estimate this quantity.

## C   Connection with K-means clustering and optimal transport

If we introduce a temperature parameter, $\tau$, fix the parameters of the feature encoder $\theta$, and let the negative pair-wise cost be the weighting function instead of the inner product, the conditional distribution becomes

$$\pi_{kj} := \pi_{\boldsymbol{\theta}}(\boldsymbol{\mu}_k | \boldsymbol{f}_j^t) = \frac{p(\boldsymbol{\mu}_k) \exp(\frac{-c(\boldsymbol{\mu}_k, \boldsymbol{f}_j^t)}{\tau})}{\sum_{k'=1}^{K} p(\boldsymbol{\mu}_{k'}) \exp(\frac{-c(\boldsymbol{\mu}_{k'}, \boldsymbol{f}_j^t)}{\tau})}. \tag{12}$$

With a uniform prior and letting $\tau \to 0$, the conditional distribution becomes a one-hot encoding, $\pi_{kj} = \mathbf{1}_{\{k=\underset{k'}{\arg\min} \, c(\boldsymbol{\mu}_{k'}, \boldsymbol{f}_j^t)\}}$, which is equivalent to solving the following constrained optimization problem:

$$\min_{\pi_{kj}} \quad \frac{1}{M} \sum_{j=1}^{M} \sum_{k=1}^{K} \pi_{kj} c(\boldsymbol{\mu}_k, \boldsymbol{f}_j^t) \tag{13}$$

$$s.t. \quad \sum_{k=1}^{K} \pi_{kj} = 1, \quad \forall j, \tag{14}$$

$$\pi_{kj} \in \{0, 1\} \quad \forall j, k, \tag{15}$$

where $\mu_k$ is fixed. This is exactly the cluster assignment step in the K-Means clustering algorithm [80]. In other words, we assign each data point to its closest centroid. Thus, the update in the cross-entropy loss can be interpreted as the cluster-center update step and the update in the transport loss is analogous to the cluster assignment step.

As explained in the main text, we might not be able to rely on the cost, $c(\boldsymbol{\mu}_k, \boldsymbol{f}_j^t)$, alone because we do not have labels in the target domain. To avoid degenerate solutions, one might consider introducing a balanced constraint: each cluster should contain an equal number of data points. The constrained optimization problem then becomes:

$$\min_{\pi_{kj}} \quad \frac{1}{M} \sum_{j=1}^{M} \sum_{k=1}^{K} \pi_{kj} c(\boldsymbol{\mu}_k, \boldsymbol{f}_j^t) \tag{16}$$

$$s.t. \quad \sum_{j=1}^{M} \pi_{kj} = \frac{M}{K}, \quad \forall k \tag{17}$$

$$\pi_{kj} \in \{0, 1\}, \quad \forall k. \tag{18}$$

This is an integer programming problem and may look difficult to optimize. However, one can relax the decision variables, $\pi_{kj}$, to be continuous and solve the following constrained optimization problem instead:

$$\min_{\pi_{kj}} \quad \frac{1}{M} \sum_{j=1}^{M} \sum_{k=1}^{K} \pi_{kj} c(\boldsymbol{\mu}_k, \boldsymbol{f}_j^t) \tag{19}$$

$$s.t. \quad \sum_{j=1}^{M} \pi_{kj} = \frac{1}{K}, \quad \forall k \tag{20}$$

$$\sum_{k=1}^{K} \pi_{kj} = \frac{1}{M}, \quad \forall j \tag{21}$$

$$\pi_{kj} \geq 0, \quad \forall j, k. \tag{22}$$

The formulation above is the optimal transport problem discussed in the text where the marginal constraints are uniform distributions over data points and classes. While we are solving a continuous relaxation of the integer programming problem, solving this problem leads to an integral solution [81], meaning that the optimal transport problem is equivalent to the cluster assignment step of the K-means algorithm with a balanced constraint. Note that the statement holds when $M$ is divisible by $K$.

## D   Implementation details

We implement our method on top of the open-source transfer learning library (MIT license) [66], adopting the default neural network architectures for both the feature encoder and linear classifier. For the feature encoder network, we utilize a pre-trained ResNet-50 in all experiments except for multi-source domain adaptation, where we use a pre-trained ResNet-101. We fine-tune the feature encoder and train the linear layers from random initialization. The linear layers have the learning rate of 0.01, ten times that of the feature encoder. The learning rate follows the following schedule: $\eta_{\text{iter}} = \eta_0 (1 + \gamma \text{iter})^{-\alpha}$, where $\eta_0$ is the initial learning rate. We set $\eta_0$ to 0.01, $\gamma$ to 0.0002, and $\alpha$ to 0.75. We utilize a mini-batch SGD with a momentum of 0.9. We set the batch size for the source data as $N = 32$ and that for the target data as $M = 96$. We use all the labeled source samples and unlabeled target samples [6, 7, 26]. We set $\beta_0$ to 0 (using a uniform prior) in all settings except for the sub-sampled target datasets. We perform a sensitivity analysis (see Appendix E) and set $\beta_0$ empirically to 0.001 for the sub-sampled target version of Office-31 and 0.0001 for that of Office-Home. We report the average accuracy from three independent runs. All experiments are conducted using a single Nvidia Tesla V100 GPU except for the DomainNet experiment, where we use four V100 GPUs.

In both the single and multi-source settings, we set $\beta_0 = 0$, which corresponds to using a uniform prior. We do not perform any additional hyper-parameter searches. The cross-entropy and transport losses are equally weighted. We run each experiment for 10,000 iterations for the single-source setting. For the multi-source setting, we train the model for 75,000 iterations.

In the class imbalance setting, we follow the experimental protocol in Tachet des Combes et al. [18] and quote the results directly when available. We perform a sensitivity analysis on the parameter, $\beta_0$, and present the result in Table 8. In the sub-sampled target datasets, we empirically set $\beta_0$ to 0.001 for Office-31 and 0.0001 for Office-Home. For sub-sampled source datasets, we set $\beta_0$ to 0 in all experiments. In all of the above settings, we run three independent experiments using the seeds $\{0, 1, 2\}$.

In the source-private domain adaptation setting, we implement our method using the same setup as Liang et al. [31]. We use all the same hyper-parameters except for the maximum number of epochs and target batch size. We set the number of epochs to 70 for Office-31 and 100 for Office-Home. The target batch size is set to 96. We change the two parameters to adjust for the number of data seen since SHOT goes through the whole training set at every 15 iterations. We use the same random seeds, $\{2019, 2020, 2021\}$, as the original paper.

# E   Additional experimental results

**Details on the synthetic experiment (Figure 2).**  In this experiment, we provide an illustration of our method as well as the baseline, DANN, on a toy dataset. We sample two dimensional data from multivariate Gaussian distributions with different means but the same covariance matrix, $\Sigma = \begin{bmatrix} 0.5 & -0.3 \\ -0.3 & 0.5 \end{bmatrix}$. In each domain, we draw $300$ examples. We draw $250$ of the green class from $\mathcal{N}([7, 5.5], \Sigma)$ and $50$ of the red class from $\mathcal{N}([4, 3.5], \Sigma)$. In the target domain, we draw $50$ of the green class from $\mathcal{N}([7.5, 3.5], \Sigma)$ and $250$ of the red class from $\mathcal{N}([4.4, 5], \Sigma)$. We utilize a three-layer feature encoder with hidden dimension $15$ and output dimension $2$ to visualize the latent space. The classifier is a linear layer.

**Visualization of transport probabilities.**  In Figure 5, we visualize transport probabilities on a sample batch of data for PCT and POT. As explained earlier, OT is equivalent to solving a balanced-constrain K-means when $M$ is divisible by $K$ so we set $M = K = 31$. In Figure 5b, we can see that OT gives equal weight to all the assigned points, and each row (class) contains only one active cell, meaning that each data point is assigned into a distinct cluster. In Figure 5a, the active cells in $\pi_{\boldsymbol{\theta}}(\boldsymbol{\mu}_k \,|\, \boldsymbol{f}_j^t)$ usually correspond to those in $\pi_{\boldsymbol{\theta}}(\boldsymbol{f}_j^t \,|\, \boldsymbol{\mu}_k)$. However, the magnitudes often differ: $\pi_{\boldsymbol{\theta}}(\boldsymbol{\mu}_k \,|\, \boldsymbol{f}_j^t)$ takes into account the uncertainty in the classes whereas $\pi_{\boldsymbol{\theta}}(\boldsymbol{f}_j^t \,|\, \boldsymbol{\mu}_k)$ considers the uncertainty of the target features. $\pi_{\boldsymbol{\theta}}(\boldsymbol{\mu}_k \,|\, \boldsymbol{f}_j^t)$ will have at least one active cell across the rows (every data point is close to some prototype), while $\pi_{\boldsymbol{\theta}}(\boldsymbol{f}_j^t \,|\, \boldsymbol{\mu}_k)$ will have at least one active cell across the columns (every prototype is close to some target feature).

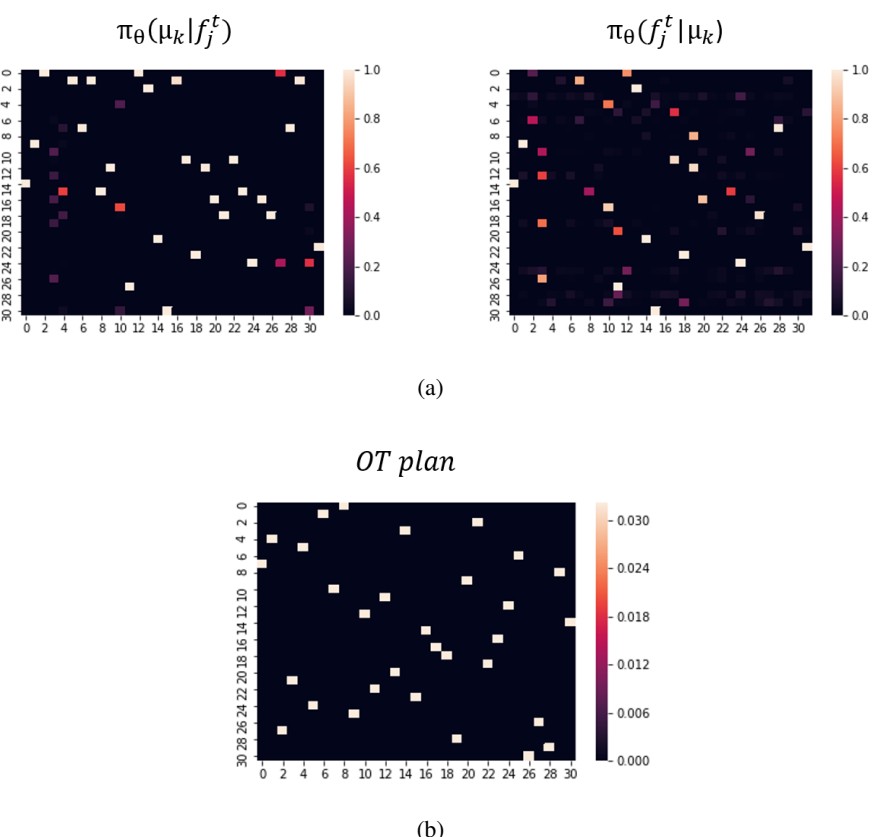

Figure 5: Visualization of transport probabilities for PCT and POT. The rows correspond to different $\boldsymbol{\mu}_k$ whereas the columns correspond to different $\boldsymbol{f}_j^t$.

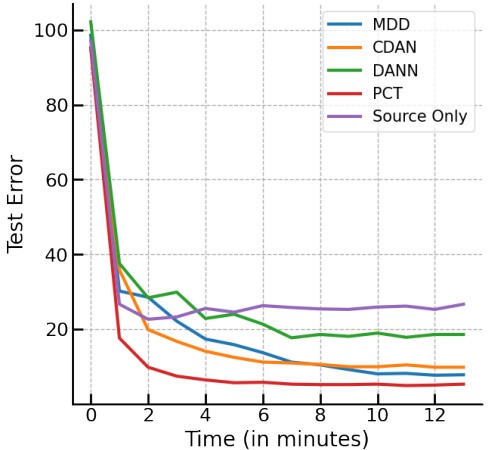

Figure 6: Test error vs. training time in minutes for different algorithms trained on the task A → W.

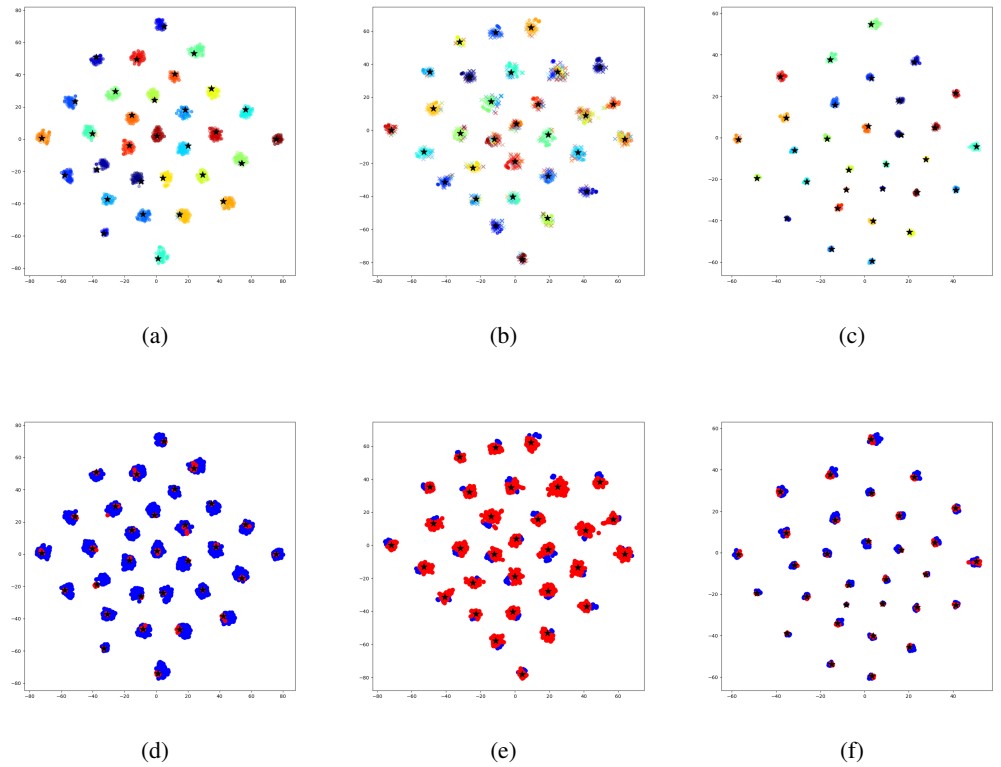

Figure 7: TSNE visualizations on the Office-31 dataset. The plots in each column correspond to each transfer task: (a),(d) to A→D, (b),(e) to D→A, and (c),(f) to W→D. Plots in the top row highlight class information whereas those in the bottom row exhibit domain information. In the top row, each plot shows that both the source (dots ·) and target (crosses ×) features are close to the prototypes (black stars ⋆). In the bottom row, we can see that the blue dots (source domain) are close to the red dots (target domain).

**Sensitivity analysis.** In Table 8, we can see that $\beta_0$ works well in the range of $0.0001 - 0.01$. Generally, $\beta_0$ should be set to a small value because the average predictions of a single mini-batch can be noisy. Thus, we give more weight to the weighted sum of the past average predictions over multiple mini-batches, which are more stable.

Table 8: Accuracy (%) on the task (A→ sW) for the sub-sampled (target) Office-31 for different values of $\beta_0$ (ResNet-50).

| $\beta_0$ | 1 | 0.1 | 0.01 | 0.001 | 0.0001 | 0 |
|---|---|---|---|---|---|---|
| | 37.8 | 80.2 | 88.5 | **88.9** | 88.5 | 87.2 |

## E.1 Full experimental results

Due to space constraints, we report the average accuracy of different transfer tasks of a dataset in some experiments. Below, we present the full tables, which include the average accuracy of the individual tasks.

### E.1.1 Ablation study

Table 9: Accuracy (%) of PCT on Office-31 under different variants (ResNet-50).

| Method | A → W | D → W | W → D | A → D | D → A | W → A | Avg |
|---|---|---|---|---|---|---|---|
| PCT w/o ($\mathcal{L}_{t \to \mu}$) | 92.7 | 98.1 | 99.8 | 92.0 | 75.3 | 73.6 | 88.6 |
| PCT w/o ($\mathcal{L}_{\mu \to t}$) | 84.4 | 98.5 | 99.9 | 89.8 | 69.0 | 64.0 | 84.3 |
| POT | 94.1 | 97.6 | 97.8 | 89.7 | 74.0 | 74.1 | 87.9 |
| POT-Sinkhorn | 94.4 | 98.1 | 98.3 | 89.6 | 75.3 | 74.0 | 88.3 |
| w/o stop-grad $\mu$ | 92.4 | 98.8 | **100.0** | 93.4 | 75.8 | 73.8 | 89.0 |
| $c(\boldsymbol{\mu}_k, \boldsymbol{f}_j^t) = \exp(-\boldsymbol{\mu}_k^T \boldsymbol{f}_j^t)$ | 90.4 | **98.9** | 100.0 | 89.9 | 72.8 | 69.5 | 86.9 |
| PCT (Ours) | **94.6** $\pm$ 0.5 | 98.7 $\pm$ 0.4 | 99.9 $\pm$ 0.1 | **93.8** $\pm$ 1.8 | **77.2** $\pm$ 0.5 | **76.0** $\pm$ 0.9 | **90.0** |

### E.1.2 Sub-sampled setting

Table 10: Accuracy (%) on the sub-sampled (source) Office-31 for unsupervised domain adaptation (ResNet-50).

| Method | sA → W | sD → W | sW → D | sA → D | sD → A | sW → A | Avg |
|---|---|---|---|---|---|---|---|
| ResNet-50 | 70.7 | 95.3 | 97.3 | 75.8 | 56.8 | 58.4 | 75.7 |
| DANN | 77.7 | 93.8 | 96.0 | 75.5 | 56.6 | 57.5 | 76.2 |
| JAN | 77.6 | 91.7 | 92.6 | 77.8 | 64.5 | 65.1 | 78.2 |
| CDAN | 84.6 | 96.8 | 98.3 | 82.5 | 62.5 | 65.0 | 81.6 |
| IWDANN | 88.4 | 97.0 | 98.7 | 81.6 | 65.0 | 64.9 | 82.6 |
| IWJAN | 83.3 | 96.3 | 98.8 | 84.6 | 65.3 | 67.4 | 82.6 |
| IWCDAN | 87.3 | 97.7 | 99.0 | 86.6 | 66.5 | 66.3 | 83.9 |
| PCT-Uniform (Ours) | **92.4** $\pm$ 1.2 | **97.8** $\pm$ 0.4 | **99.4** $\pm$ 0.0 | **91.1** $\pm$ 2.2 | **73.9** $\pm$ 0.5 | **73.0** $\pm$ 1.1 | **87.9** |

Table 11: Accuracy (%) on the sub-sampled (target) Office-31 for unsupervised domain adaptation (ResNet-50).

| Method | A → sW | D → sW | W → sD | A → sD | D → sA | W → sA | Avg |
|---|---|---|---|---|---|---|---|
| ResNet-50 | 68.4 | 96.7 | 99.3 | 68.9 | 62.5 | 60.7 | 76.1 |
| DANN | 76.3 | 88.0 | 93.0 | 72.9 | 62.3 | 63.1 | 75.9 |
| JAN | 78.5 | 89 | 92.1 | 81.4 | 62.9 | 64.9 | 78.1 |
| CDAN | 85.8 | 97.6 | 99.9 | 85.2 | 64.9 | 64.6 | 83.0 |
| IWDANN | 76.4 | 97.1 | **100.0** | 82.7 | 59.0 | 59.9 | 79.2 |
| IWJAN | 83.6 | 97.9 | 99.7 | 86.2 | 64.0 | 65.6 | 82.8 |
| IWCDAN | 87.9 | 97.7 | **100.0** | 86.2 | 64.8 | 64.1 | 83.5 |
| PCT-Uniform (Ours) | 86.4 $\pm$ 0.86 | 97.3 $\pm$ 0.3 | **100.0** $\pm$ 0.5 | 88.5 $\pm$ 0.8 | 69.5 $\pm$ 0.9 | 70.5 $\pm$ 0.7 | 85.4 |
| PCT-Learnable (Ours) | **88.1** $\pm$ 0.5 | **98.5** $\pm$ 0.4 | 99.9 $\pm$ 0.17 | **90.4** $\pm$ 1.7 | **69.9** $\pm$ 0.37 | **71.3** $\pm$ 0.51 | **86.4** |

Table 12: Accuracy (%) on the sub-sampled (source) Office-Home for unsupervised domain adaptation (ResNet-50).

| Method | sAr→Cl | sAr→Pr | sAr→Rw | sCl→Ar | sCl→Pr | sCl→Rw | sPr→Ar | sPr→Cl | sPr→Rw | sRw→Ar | sRw→Cl | sRw→Pr | Avg |
|---|---|---|---|---|---|---|---|---|---|---|---|---|---|
| ResNet-50 | 35.7 | 54.7 | 62.6 | 43.7 | 52.5 | 56.6 | 44.3 | 33.0 | 65.2 | 57.1 | 40.5 | 70.0 | 51.4 |
| DANN | 36.1 | 54.2 | 61.7 | 44.3 | 52.6 | 56.4 | 44.6 | 37.1 | 65.2 | 56.7 | 43.2 | 69.9 | 51.8 |
| JAN | 34.5 | 56.9 | 64.5 | 46.2 | 56.8 | 59.0 | 50.6 | 37.2 | 70.0 | 58.7 | 40.6 | 72.00 | 53.9 |
| CDAN | 38.9 | 56.8 | 64.8 | 48.0 | 60.0 | 61.2 | 49.7 | 41.4 | 70.2 | 62.4 | 47.0 | 74.7 | 56.3 |
| IWDANN | 39.8 | 63.0 | 68.7 | 47.4 | 61.1 | 60.4 | 50.4 | 41.6 | 72.5 | 61.0 | 49.4 | 76.1 | 57.6 |
| IWJAN | 36.2 | 61.0 | 66.3 | 48.7 | 59.9 | 61.9 | 52.9 | 37.7 | 70.9 | 60.3 | 41.5 | 73.3 | 55.9 |
| IWCDAN | 43.0 | 65.0 | 71.3 | 52.9 | 64.7 | 66.5 | 54.9 | 44.8 | 75.9 | 67.0 | 50.5 | 78.6 | 61.2 |
| PCTUniform (Ours) | **51.9** ± 0.2 | **69.7** ± 0.9 | **76.5** ± 0.3 | **63.3** ± 1.3 | **70.8** ± 0.4 | **71.1** ± 0.5 | **66.0** ± 0.8 | **49.9** ± 0.7 | **80.2** ± 0.5 | **73.1** ± 0.6 | **58.6** ± 0.7 | **83.2** ± 0.8 | **67.8** |

Table 13: Accuracy (%) on the sub-sampled (target) Office-Home for unsupervised domain adaptation (ResNet-50).

| Method | Ar→sCl | Ar→sPr | Ar→sRw | Cl→sAr | Cl→sPr | Cl→sRw | Pr→sAr | Pr→sCl | Pr→sRw | Rw→sAr | Rw→sCl | Rw→sPr | Avg |
|---|---|---|---|---|---|---|---|---|---|---|---|---|---|
| ResNet-50 | 41.5 | 65.8 | 73.6 | 52.2 | 59.5 | 63.6 | 51.5 | 36.4 | 71.3 | 65.2 | 42.8 | 75.4 | 58.2 |
| DANN | 47.8 | 55.9 | 66.0 | 45.3 | 54.8 | 56.8 | 49.4 | 48.0 | 70.2 | 65.4 | 55.5 | 72.7 | 58.3 |
| JAN | 45.8 | 69.7 | 74.9 | 53.9 | 63.2 | 65.0 | 56 | 42.5 | 74 | 65.9 | 47.4 | 78.8 | 61.4 |
| CDAN | 51.1 | 69.7 | 74.6 | 56.9 | 60.4 | 64.6 | 57.2 | 45.5 | 75.6 | 68.5 | 52.7 | 79.8 | 63.0 |
| IWDANN | 48.7 | 62.0 | 71.6 | 50.4 | 57.0 | 60.3 | 51.4 | 41.1 | 69.9 | 62.6 | 51.0 | 77.2 | 58.6 |
| IWJAN | 44.0 | 71.9 | 75.1 | 55.2 | 65.0 | 67.7 | 57.1 | 42.4 | 74.9 | 66.1 | 46.1 | 78.5 | 62.0 |
| IWCDAN | 52.3 | 72.2 | 76.3 | 56.9 | 67.3 | 67.7 | 57.2 | 46.0 | 77.8 | 67.3 | 53.8 | 80.6 | 64.6 |
| PCT-Uniform (Ours) | 55.8 ± 0.5 | 77.6 ± 0.6 | 80.4 ± 0.3 | 65.1 ± 1.2 | 72.3 ± 2.0 | 74.7 ± 0.2 | **67.0** ± 1.5 | 50.9 ± 1.0 | 81.1 ± 0.3 | 72.6 ± 0.2 | 57.0 ± 0.2 | **84.0** ± 0.2 | 69.8 |
| PCT-Learnable (Ours) | **57.5** ± 0.4 | **78.2** ± 0.2 | **80.5** ± 0.0 | **66.7** ± 0.6 | **74.3** ± 1.3 | **75.4** ± 0.5 | 64.6 ± 1.5 | 50.7 ± 1.4 | **81.3** ± 0.4 | **72.9** ± 0.3 | **57.3** ± 0.9 | 83.5 ± 0.15 | **70.2** |

### E.1.3 Source-private setting

Table 14: Accuracy (%) on the source-private Office-31 for unsupervised domain adaptation (ResNet-50).

| Method | A → W | D → W | W → D | A → D | D → A | W → A | Avg |
|---|---|---|---|---|---|---|---|
| Source Model Only | 76.8 | 95.3 | 98.7 | 80.8 | 60.3 | 63.6 | 79.3 |
| SHOT-Psuedo-Label | 90.8 | 96.6 | 99.3 | 93.2 | 72.1 | 73.5 | 87.6 |
| SHOT-IM | 91.2 | 98.3 | 99.9 | 90.6 | 72.5 | 71.4 | 87.3 |
| SHOT | 90.1 | **98.4** | **99.9** | **94.0** | **74.7** | 74.3 | **88.6** |
| PCT (Ours) | **91.7** ± 0.8 | 97.9 ± 0.3 | **99.9** ± 0.2 | 92.2 ± 1.1 | 74.0 ± 1.6 | **74.6** ± 0.3 | 88.4 |

Table 15: Accuracy (%) on the source-private Office-Home for unsupervised domain adaptation (ResNet-50).

| Method | Ar → Cl | Ar → Pr | Ar → Rw | Cl → Ar | Cl → Pr | Cl → Rw | Pr → Ar | Pr → Cl | Pr → Rw | Rw → Ar | Rw → Cl | Rw → Pr | Avg |
|---|---|---|---|---|---|---|---|---|---|---|---|---|---|
| ResNet-50 | 44.6 | 67.3 | 74.8 | 52.7 | 62.7 | 64.8 | 53.0 | 40.6 | 73.2 | 65.3 | 45.4 | 78.0 | 60.2 |
| SHOT | **57.1** | **78.1** | **81.5** | 68.0 | **78.2** | **78.1** | **67.4** | 54.9 | **82.2** | 73.3 | 58.8 | **84.3** | **71.8** |
| PCT (Ours) | 56.6 ± 1.1 | 77.0 ± 0.6 | 79.8 ± 0.5 | **68.3** ± 0.5 | 75.7 ± 0.4 | 75.5 ± 0.3 | 67.3 ± 1.3 | **55.1** ± 1.0 | 80.2 ± 0.9 | **74.4** ± 0.5 | **58.9** ± 0.5 | 83.2 ± 0.7 | 71.0 |

## E.2 Additional Results

To further verify the efficacy of our framework, we provide additional results on the Cross-Digits, Office-Caltech, Image-Clef, and VisDA under different settings.

### E.2.1 Single-source setting

Table 16: Average Accuracy (%) on the Cross-Digits dataset for unsupervised domain adaptation (ResNet-50).

| Method | MNIST → USPS | SVHN → MNIST | USPS → MNIST | Avg |
|---|---|---|---|---|
| CDAN | 95.6 | 89.2 | **98.0** | 94.3 |
| rRevGrad+CAT | 94 | 98.8 | 96 | 96.3 |
| ETD | 96.4 | 97.9 | 96.3 | 96.9 |
| PCT (Ours) | **97.8** ± 0.1 | **98.9** ± 0.0 | 97.0 ± 0.6 | **98.0** |

Table 17: Average Accuracy (%) on the Office-Caltech dataset for unsupervised domain adaptation (ResNet-50).

| Method | A → W | A→D | A→C | D→A | D→W | D→C | W→A | W→D | W→C | C→A | C→W | C→D | Avg |
|---|---|---|---|---|---|---|---|---|---|---|---|---|---|
| CDAN | **99.3** | 96.8 | 95.4 | 94.7 | **100.0** | 94.6 | 95.7 | **100.0** | 94.5 | 94.8 | 95.9 | 92.4 | 96.2 |
| MDD | 98.3 | 98.0 | 94.8 | 95.3 | 98.6 | 94.3 | 95.6 | **100.0** | 94.9 | **95.8** | 96.3 | **98.7** | 96.7 |
| PCT (Ours) | 99.1 ± 0.1 | **98.5** ± 0.3 | **95.6** ± 0.1 | **96.3** ± 0.3 | 99.8 ± 0.17 | **95.1** ± 0.3 | **96.2** ± 0.1 | 100 ± 0.0 | 95.2 ± 0.2 | **95.8** ± 0.4 | **98.7** ± 0.4 | 96.6 ± 1.0 | **97.3** |

Table 18: Average Accuracy (%) on the Image-Clef dataset for unsupervised domain adaptation (ResNet-50).

| Method | I→P | P→I | I→C | C→I | C→P | P→C | Avg |
|---|---|---|---|---|---|---|---|
| CDAN | 77.7 | 90.7 | 97.7 | 91.3 | 74.2 | 94.3 | 87.7 |
| rRevGrad+CAT | 77.2 | 91 | 95.5 | 91.3 | 75.3 | 93.6 | 87.3 |
| ETD | **81** | 91.7 | **97.9** | **93.3** | **79.5** | 95 | **89.7** |
| PCT (Ours) | 78.5 ± 0.4 | **93.1** ± 0.2 | 97.0 ± 0.3 | 92.2 ± 0.2 | 75.7 ± 0.6 | **95.4** ± 0.4 | 88.7 |

### E.2.2 Multi-source setting

Table 19: Average Accuracy (%) on the Office-31 dataset for ResNet50-based MSDA methods.

| Category | Method | $\mathcal{R} \to D$ | $\mathcal{R} \to W$ | $\mathcal{R} \to A$ | Avg |
|---|---|---|---|---|---|
| Multi-source | DCTN | 99.3 | 98.2 | 64.2 | 87.2 |
| | MFSAN | 99.5 | **98.5** | 72.7 | 90.2 |
| | SImpAl | 99.2 | 97.4 | 70.6 | 89.0 |
| Source-combined | DAN | 99.6 | 97.8 | 67.6 | 88.3 |
| | D-CORAL | 99.3 | 98.0 | 67.1 | 88.1 |
| | RevGrad | 99.7 | 98.1 | 67.6 | 88.5 |
| | PCT (Ours) | **99.8** ± 0.0 | **98.5** ± 0.1 | **76.9** ± 0.6 | **91.7** |

### E.2.3 Source-private setting

Table 20: Accuracy (%) on the VisDA-2017 dataset for unsupervised domain adaptation (ResNet-50).

| ETN | STA | UAN | DANCE | PCT (Ours) |
|---|---|---|---|---|
| 64.1 | 48.1 | 66.4 | 70.2 | **71.2** ± 0.8 |