# OpenReview forum: "A Prototype-Oriented Framework for Unsupervised Domain Adaptation"
_NeurIPS.cc/2021/Conference — NeurIPS 2021 Poster_

### Official Review · Reviewer_ZA7J · 2021-07-15

**Rating:** 4
**Confidence:** 5

**Summary:**

In this paper, a UDA algorithm is proposed. The core idea is based on benefiting from the class prototypes in the source domain to align two domains in a shared embedding space, rather than using the source sample features. The class prototypes are learned on the source domain using a cross-entropy loss to relax computational load. Upon learning the prototypes, a transport-based loss function is used to align the target domain features with the learned prototypes. Experiments on several UDA scenarios using standard benchmark datasets are provided to demonstrate that the method is effective.

**Limitations And Societal Impact:**

The authors have explored these aspects in a short "broader impact" section.

**Main Review:**

1. UDA is a highly explored area and new contributions should introduce novel ideas that can address existing challenges or boost state-of-the-art performances significantly. More specifically, the idea of using prototypes for UDA has been explored extensively in the literature. Transport-based optimization also has been used extensively for this purpose. Source-free UDA algorithms based on using prototypes have been developed as well, e.g.,:

- Universal domain adaptation through self-supervision, NeurIPS 2020

- Unsupervised model adaptation for continual semantic segmentation, AAAI 2021

It is true the proposed algorithm is different in the idea implementation details but the broad solution is not novel due to the precedent. Performance results also do not demonstrate a significant boost over prior works. Finally, there is no theoretical analysis to improve our understanding about using a similar approach to address UDA. Hence, I think this work is not suitable for publication at NeurIPS.

2. Recent UDA algorithms perform more extensive experiments to demonstrate the generalizability of the method and offer benefits over prior works. In this aspect, I think the experiments need significant improvement:

- Results on some important benchmarks, including, ImageCLEF, cross-domain digits, Office-Caltech, etc, are missing.

- Multi-source and source-private experiments are even less extensive in terms of both used datasets and comparison with prior works.

- Comparison has been done against mostly methods that have not been developed in the past three years. Please check the literature to include the existing state-of-the-art performance.

- An ablative study on Eq (7) is helpful to determine the relative importance of the proposed terms.

-In lines 49-55, it is claimed that the proposed approach is advantageous compared to adversarial learning because of robustness and faster convergence properties. Experiments are based on using test error versus learning iteration to demonstrate this claim. But I think this is not fully convincing. Comparisons should be performed based on time because the algorithms are different in terms of computations at each iteration. In other words, the execution time per iteration may be very different among these methods. A second set of experiments to demonstrate robustness is going to be very helpful.


-----------------------------------
After more elaboration, I finalized my updated rating and decided to maintain it. I think this work is best suited for future venues after incorporating all the review feedback.

**Time Spent Reviewing:**

2

---

> ### Author Response · Authors · 2021-08-09
> **Point-by-point Response to Reviewer ZA7J**
>
> Thank you for your comments. We provide our point-by-point response below.
>
>
>
>
> >*UDA is a highly explored area and new contributions should introduce novel ideas.*
>
>
>
> - We believe how we utilize prototypes with the bi-directional transport loss while also estimating class proportions is a novel contribution (as also pointed out by reviewer KNFV). Our unified prototype-oriented framework can handle class imbalance while being computationally efficient and privacy-preserving. By contrast, previously proposed algorithms typically deal with these important challenges separately.
>
>
>
> >*Transport-based optimization also has been used extensively in the literature.*
>
>
>
> - While optimal transport is frequently used in domain adaptation, to the best of our knowledge, bi-directional transport loss has not been explored. In the related work section, we mention low computational complexity as an important advantage over optimal transport. In the ablation study, we also further show that applying bi-directional transport loss with prototypes yields better performance than using optimal transport with prototypes. Moreover, our bi-directional transport loss is extended to deal with class imbalance. Conventional optimal transport loss will suffer from this setting as solving the vanilla optimal transport problem is closely connected with solving a balanced clustering problem (Asano, ICLR 2020).
>
>
>
> >*Source-free UDA algorithms based on using prototypes have been developed.*
>
>
>
> - While our approach shares some similarities with Saito et al. (NeurIPS, 2020), there are major distinctions that differentiate our paper from theirs. First, they focus on universal domain adaptation (different category shift scenarios) whereas our framework addresses class imbalance, multi-source, and source-private settings in closed-set domain adaptation. These different settings we examine are salient in applications. While methods developed for universal domain adaptation can also work for closed-set domain adaptation, the performance achieved is lower compared to ours. In this setting, our method achieves 4.5 % higher accuracy in the Office-31 dataset and 2.7% in the Office-home dataset. (These tasks are the only two where the settings are the same.) Unlike Saito et al. (NeurIPS, 2020), we do not rely on domain-specific batch normalization to boost the performance. Second, our approach is much more computationally efficient since their method uses a memory bank to store the target features for the whole dataset. Their loss also needs to be computed using the whole dataset.
>
>
>
> - While Stan et al. (AAAI, 2021) also utilize prototypes in their formulation, they address a segmentation problem that is clearly different from ours. Unlike their method, our prototype construction does not require additional parameters. Moreover, they use an arg-max strategy for the matching, while we use the conditional probabilities to determine how the samples and prototypes are matched.
>
>
>
> >*Performance results also do not demonstrate a significant boost over prior works.*
>
> - In the class-imbalance setting, we compare with state-of-the-art algorithm, IWCDAN. We perform significantly better than that method, which is published in NeurIPS 2020 (4% better in the Office-31 and 6% better in the Office-Home). As pointed out earlier, in the source-free setting, our method achieves 4.5% higher accuracy in the Office-31 dataset and 2.7% in the Office-home dataset than DANCE (Saito et al., NeurIPS 2020). In the single-source setting, we also outperform a variety of baseline methods published within the past three years (MDD - ICML 2019, TPN - CVPR 2019, ETD - CVPR 2020).
>
>
>
> >*There is no theoretical analysis to improve our understanding about using a similar approach to address UDA.*
>
>
>
> - To help understand our method, we have made theoretical connections with other methods (clustering, optimal transport, and entropy minimization). In addition, we have provided extensive empirical support to corroborate our claims and understand our method:
>
>     1.  Experiments across multiple challenging problems in the closed UDA setting (single-source, multi-source, class imbalance, and source-private) (Table 1-5).
>
>   2.  Class proportion estimation for the imbalanced dataset (Appendix E, Figure 4).
>
>   3.  Visualization of latent features and the learned prototypes (Figure 3).
>
>   4.  Ablation study to understand each loss term (Table 6).
>
>   5.  Synthetic data example (Figure 2).

---

> > ### Author Response · Authors · 2021-08-09
> > **Additional experimental results to further verify the efficacy of PCT**
> >
> >
> >
> > >*I think the experiments need significant improvement. Results on some important benchmarks, including, ImageCLEF, cross-domain digits, Office-Caltech, etc, are missing.*
> >
> >
> >
> > - To demonstrate the general applicability of our method, we have already performed extensive experiments across four different domain adaptation settings: single-source, multi-source, class-imbalance, and source-private domain adaptation.
> >
> >
> >
> > - To address your concerns, we have run our algorithm on ImageCLEF, cross-domain digits, and Office-Caltech, which are shown below and will also be included in the revision. These results further verify the efficacy of the proposed PCT.
> >
> >
> >
> >    - ### Cross digit
> >
> >
> >      |                                  | MNIST -> USPS | SVHN -> MNIST | USPS -> MNIST | Avg acc |
> > | -------------------------------- | ------------- | ------------- | ------------- | ------- |
> > | CDAN, (Long, Neurips, 2018)      | 95.6          | 89.2          | **98.0**          | 94.3    |
> > | rRevGrad+CAT, (Deng, ICCV, 2019) | 94            | 98.8          | 96            | 96.3    |
> > | ETD, (Li, CVPR, 2020)            | 96.4          | 97.9          | 96.3          | 96.9    |
> > | PCT (Ours)                       | **97.8** ± 0.1    | **98.9** ± 0.0    | 97.0 ± 0.6    | **98.0**    |
> >
> >
> >
> > - ### OfficeCaltech
> >
> >
> >    |                             | A2W        | A2D        | A2C        | D2A        | D2W         | D2C        | W2A        | W2D       | W2C        | C2A        | C2W        | C2D        | Avg acc |
> > | --------------------------- | ---------- | ---------- | ---------- | ---------- | ----------- | ---------- | ---------- | --------- | ---------- | ---------- | ---------- | ---------- | ------- |
> > | CDAN, (Long, NeurIPS, 2018) | **99.3**       | 96.8       | 95.4       | 94.7       | **100.0**       | 94.6       | 95.7       | **100.0**     | 94.5       | 94.8       | 95.9       | 92.4       | 96.2    |
> > | MDD, (Zhang, ICML, 2019)    | 98.3       | 98.0       | 94.8       | 95.3       | 98.6        | 94.3       | 95.6       | **100.0**     | 94.9       | **95.8**      | 96.3       | **98.7**       | 96.7    |
> > | PCT (Ours)                  | 99.1 ± 0.1 | **98.5** ± 0.3 | **95.6** ± 0.1 | **96.3** ± 0.3 | 99.8 ± 0.17 | **95.1** ± 0.3 | **96.2** ± 0.1 | **100** ± 0.0 | **95.2** ± 0.2 | **95.8** ± 0.4 | **98.7** ± 0.4 | 96.6 ± 1.0 | **97.3**   |
> >
> >
> >
> >  - ### ImageClef
> >
> >
> > |                                  | I2P        | P2I        | I2C        | C2I        | C2P        | P2C        | Avg acc |
> > | -------------------------------- | ---------- | ---------- | ---------- | ---------- | ---------- | ---------- | ------- |
> > | CDAN, (Long, Neurips, 2018)      | 77.7       | 90.7       | 97.7       | 91.3       | 74.2       | 94.3       | 87.7    |
> > | rRevGrad+CAT, (Deng, ICCV, 2019) | 77.2       | 91         | 95.5       | 91.3       | 75.3       | 93.6       | 87.3    |
> > | ETD, (Li, CVPR, 2020)            | **81**         | 91.7       | **97.9**       | **93.3**       | **79.5**       | 95         | **89.7**    |
> > | PCT (Ours)                       | 78.5 ± 0.4 | **93.1** ± 0.2 | 97.0 ± 0.3 | 92.2 ± 0.2 | 75.7 ± 0.6 | **95.4** ± 0.4 | 88.7    |
> >
> >
> >
> >
> > **_NOTE:_** Even though ETD performs 1% better than PCT on ImageClef, PCT is better than ETD by 3.8% on Office-31 and by 4.5% on Office-Home (Table 1 in the paper). ETD also introduces an additional attention network to compute the transport costs. Unlike ETD, PCT does not require additional parameters and maintains better efficiency.
> >
> >
> >
> > For experiments on ImageClef and OfficeCaltech, we use version 1 of the thuml/Transfer-Learning-Library code base in Github.
> >
> > For experiments on cross-digit dataset, we use the tim-learn/SHOT code base in Github.
> >
> >
> >
> > $~$
> >
> > >*Multi-source and source-private experiments are even less extensive in terms of both used datasets and comparison with prior works.*
> >
> >
> >
> > - As we try to demonstrate our method on four different settings (single-source, multi-source, class imbalance, and source private), we provide one to two representative experiments for each setting. In the multi-source setting, we test our method on a commonly used large-scale dataset, DomainNet (569,010 images with 345 categories). Additional results on the Office-Home dataset are also included in the appendix.
> >
> >
> >
> > - To further address your concerns, we have added two more multi-source experiments (Office-31 and Office-Home) and one more source-private experiment (Visda-C).
> >
> >
> > - ### Multi-source, Office-31
> >
> >
> > |                 |                                   | Rest -> D | Rest -> W | Rest -> A | Avg  |
> > | --------------- | --------------------------------- | --------- | --------- | --------- | ---- |
> > |                 | DCTN, (Xu, CVPR, 2018)            | 99.3      | 98.2      | 64.2      | 87.2 |
> > | Multi-Source    | MFSAN, (Zhu, AAAI, 2019)          | 99.5      | **98.5**      | 72.7      | 90.2 |
> > |                 | SImpAl50, (Venkat, NeurIPS, 2020) | 99.2      | 97.4      | 70.6      | 89.0   |
> > |                 |                                   |           |           |           |      |
> > | Source-Combined | PCT (Ours)                        | **99.8**      | **98.5**      | **76.9**     | **91.7** |
> >
> >
> >
> > - ### Multi-source, Office-Home
> >
> > |                 |                                   | Rest -> Ar | Rest -> Cl | Rest -> Pr | Rest -> Rw | Avg  |
> > | --------------- | --------------------------------- | ---------- | ---------- | ---------- | ---------- | ---- |
> > | Multi-Source    | MFSAN, (Zhu, AAAI, 2019)          | 72.1       | 62         | 80.3       | 81.8       | 74.1 |
> > |                 | SImpAI50, (Venkat, NeurIPS, 2020) | 70.8       | 56.3       | 80.2       | 81.5       | 72.2 |
> > |                 |                                   |            |            |            |            |      |
> > | Source-Combined | PCT (Ours)                        | **76.3**       | **64.1**       | **84.9**       | **84.3**       | **77.4** |
> >
> >
> >
> >
> > - ### Source-private, Visda
> >
> > | Visda (Resnet50)             | Accuracy |
> > | ---------------------------- | -------- |
> > | ETN (Cao, CVPR, 2019)        | 64.1     |
> > | STA (Liu, CVPR, 2019)        | 48.1     |
> > | UAN (Kaichao, CVPR, 2019)    | 66.4     |
> > | DANCE (Saito, NeurIPS, 2020) | 70.2     |
> > | PCT (Ours)                   | **71.2**    |
> >
> > **_NOTE:_** For multi-source experiments, the baselines are taken from Venkat et al. (NeurIPS 2020). For source-private experiments, the baselines are taken from Saito et al. (NeurIPS 2020).
> >
> > $~$
> >
> > >*Comparison has been done against mostly methods that have not been developed in the past three years.*
> >
> >
> >
> >
> > - We respectfully disagree with this comment. While we did include some classical baselines, such as DAN and CDAN, we have also compared with a number of methods published within the last three years: MDD - ICML 2019, TPN - CVPR 2019, ETD - CVPR 2020, ML-MSDA - 2020, SHOT - ICML 2020, IWCDAN - NeurIPS 2020.
> >
> >
> >
> > >*An ablative study on Eq (7) is helpful to determine the relative importance of the proposed terms.*
> >
> >
> >
> > - We would like to clarify we have already provided an ablation study of Eq (7) as well as other components in Section 4.2 in Table 6.
> >
> >
> >
> > >*Comparisons should be performed based on time because the algorithms are different in terms of computations at each iteration.*
> >
> >
> >
> > - We would like to clarify that we have provided such a figure (test accuracy vs. time) in Figure 5 in Appendix E. This figure shows that our approach still converges faster while achieving higher accuracy. We can swap the places of this figure with Figure 3b in the main paper if desired.
> >
> >   $~$
> >
> > **References:**
> >
> >
> >
> > Asano, Y.M., Rupprecht, C. and Vedaldi, A. Self-labelling via simultaneous clustering and representation learning. ICLR, 2020.
> >
> >
> >
> > Cao, Z., You, K., Long, M., Wang, J., & Yang, Q. Learning to transfer examples for partial domain adaptation. CVPR, 2019.
> >
> >
> >
> > Deng, Z., Luo, Y., & Zhu, J. Cluster alignment with a teacher for unsupervised domain adaptation. ICCV, 2019.
> >
> >
> >
> > Li, M., Zhai, Y. M., Luo, Y. W., Ge, P. F., & Ren, C. X. Enhanced transport distance for unsupervised domain adaptation. CVPR, 2020.
> >
> >
> >
> > Liu, H., Cao, Z., Long, M., Wang, J., & Yang, Q. Separate to adapt: Open set domain adaptation via progressive separation. CVPR, 2019.
> >
> >
> >
> > Long, M., Cao, Z., Wang, J., & Jordan, M. I. Conditional adversarial domain adaptation. NeurIPS, 2018.
> >
> >
> >
> > Saito, K., Kim, D., Sclaroff, S., & Saenko, K. Universal domain adaptation through self supervision. arXiv preprint arXiv:2002.07953. NeurIPS, 2020.
> >
> >
> >
> > Stan, S., & Rostami, M. Unsupervised model adaptation for continual semantic segmentation. AAAI, 2021.
> >
> >
> >
> > Venkat, N., Kundu, J. N., Singh, D. K., Revanur, A., & Babu, R. V. Your Classifier can Secretly Suffice Multi-Source Domain Adaptation. NeurIPS, 2020.
> >
> >
> >
> > Xu, R., Chen, Z., Zuo, W., Yan, J., & Lin, L. Deep cocktail network: Multi-source unsupervised domain adaptation with category shift. CVPR, 2018.
> >
> >
> >
> > You, K., Long, M., Cao, Z., Wang, J., & Jordan, M. I. (2019). Universal domain adaptation. CVPR, 2019.
> >
> >
> >
> >
> > Zhang, Y., Liu, T., Long, M., & Jordan, M. Bridging theory and algorithm for domain adaptation. ICML, 2019.
> >
> >
> >
> > Zhu, Y., Zhuang, F., & Wang, D. Aligning domain-specific distribution and classifier for cross-domain classification from multiple sources. AAAI, 2019.

---

> > > ### Comment · Reviewer_ZA7J · 2021-08-17
> > > **Update**
> > >
> > > Dear authors,
> > > Thank you for your response. I increased my score but I still think the contribution of this does not rise to NeurIPS level. Please my responses below.
> > >
> > > I agree that your work is novel in that it is a new UDA algorithm. But my concern is that there are many works in the field. I can list about 10 papers on source-free UDA. Hence, you need to demonstrate that either you address a challenge that existing works do not address or much better results on existing baselines. Your new results indicate that the proposed algorithm does not lead to a significant boost and sometimes prior works outperform it.
> > >
> > > Finally, I appreciate these new results but my concern is that the accepted version of the paper might not be a substantially improved version of the paper. The rebuttal stage is not similar to a journal revision where you can submit an improved version of your paper. Rather, it should be used to clarify misunderstandings. Hence, I base my judgment based on the submitted version of your work.

---

> > > > ### Author Response · Authors · 2021-08-17
> > > > **Clarifications**
> > > >
> > > > Dear Reviewer ZA7J,
> > > >
> > > > We appreciate your providing additional feedback and raising your score. We hope our response below could help convince you to reconsider the significance of our contributions.
> > > >
> > > > 1. We appreciate your recognition of the novelty of the proposed PCT algorithm for UDA.
> > > >
> > > > 2. We’d like to emphasize the proposed PCT is not a method restricted to the source-private (also known as source-free) UDA setting, instead, it is a general prototype-oriented framework that can handle single-source, multi-source, source-private, and class-imbalance settings, while being computationally efficient.
> > > >
> > > > 3. A challenge that has not been addressed before is constructing a computationally-efficient framework that can readily work for a variety of different application scenarios for UDA, providing state-of-the-art performance while having no need to separately reconfigure the model for each scenario.
> > > >
> > > > 4. Our new results show that the proposed PCT outperforms the state-of-the-art in almost all datasets under a variety of different application scenarios. The only exception is observed on the ImageClef dataset under the single-source UDA setting, where ETD outperforms PCT by 1 %. However, we’d like to emphasize that PCT is better than ETD by 3.8% on Office-31 and by 4.5% on Office-Home (Table 1 in the paper). These results suggest that PCT overall has a better performance than ETD. We also note ETD introduces an additional attention network, while PCT does not require additional parameters and maintains better efficiency.
> > > > Our results show overall better performance than the baselines on the datasets you suggested. We also demonstrate that our algorithm is computationally more efficient than existing approaches both in terms of time and memory costs (Figure 3a and 3b in the paper).
> > > >
> > > > 5. The key reason we ran these additional experiments and included the results into the rebuttal was not to further strengthen the experiments but to address your doubts on them.
> > > >
> > > >      - More results will always help. However, given the page constraint and as the current paper has already had a large number of experimental results to sufficiently validate our contributions, we actually don’t think it is that necessary to include these new results into the revision. As other reviewers have pointed out, our experiments are extensive (Reviewer KNFV), the analysis in Section 4 (and Appendix) is comprehensive (Reviewer Z3WR), and the paper contains well-supported experiments (Reviewer JCFS).
> > > >
> > > >     - With that being said, as these new results do provide additional evidence to further validate our contributions, we plan to add them into the Appendix.
> > > >
> > > >     - We believe adding these new results into the Appendix will help further strengthen the paper. However, the significance of doing that, in our opinion, is far from “substantial improvement” as it mainly helps provide reassurance about the efficacy and versatility of the proposed method.

---

> ### Author Response · Authors · 2021-08-29
> **Thank you for explaining your rating**
>
> Dear Reviewer ZA7J,
>
> We sincerely appreciate you responding to our rebuttal and explaining your rating. We will carefully revise our paper to incorporate these valuable comments and suggestions of all four reviewers. We believe our revisions will mainly help enhance the clarity and provide further reassurance on both the efficacy and versatility of the proposed prototype-oriented framework. In our opinion, these revisions, which are common and often expected for machine learning conference submissions, do not count as a convincing reason against acceptance.

---

### Official Review · Reviewer_JCfs · 2021-07-16

**Rating:** 6
**Confidence:** 3

**Summary:**

This paper deals with the problem of unsupervised domain adaptation (UDA) based on a prototype-oriented framework which is suggested to solve the fundamental issues on UDA (e.g., sampling variability in the source domain, instance class-mismatching in batch, source data privacy). The main contribution suggested in this paper is utilizing the linear classifier's weights as class prototypes with bi-directional transport loss for feature alignment between source and target domains.

**Limitations And Societal Impact:**

I am curious what kind of limitations that this paper could have when compared with other prototype-based alignment papers.

**Main Review:**

Originality:

 As suggested in the main contribution, the author claims the originality of using learning parameters for class prototypes and bidirectional prototype-oriented transport loss.

When it comes to constructing class prototypes with learnable parameters, I doubt the originality since the below paper also utilizes the learnable parameters to construct the class prototype for labeled data.

- Semi-supervised Domain Adaptation via Minimax Entropy, Saito et al.

It would be more convincing if the author can prove the originality and show the main difference from the above paper.

Quality:

The motivation of using learnable parameters for class prototype and technical approach using prototype-oriented transport loss seems to be reasonable.  It is supported with various unsupervised domain adaptation settings (one source, multi-source, source-data-private). Furthermore, explicit comparisons on the number of parameters and converging speed prove its effectiveness on practical domain adaptation.

However, I have one concern related to the quality of this paper.
Comparison with proposed baselines seems to be insufficient. Specifically, the proposed method needs to be compared with other prototype-based alignment methods.

- Contrastive Adaptation Network for Unsupervised Domain Adaptation, Kang et al.
- Prototypical Cross-domain Self-supervised Learning for Few-shot Unsupervised Domain Adaptation, Yue et al.

Clarity:

It is easy to follow and well-written. The paper contains well-supported theoretical analysis and experiments.

Significance:

Suggesting the framework that requires no additional model parameters and converges fast seems to be applied in real-world domain adaptation problems.


**Time Spent Reviewing:**

More than one day

---

> ### Author Response · Authors · 2021-08-09
> **Point-by-point Response to the comments of Reviewer JCfs**
>
> Thank you for thoroughly going through our paper and providing constructive comments. Below please find our point-by-point response to your comments.
>
>
> >*It would be more convincing if the author can prove the originality and show the main difference from the above paper. Semi-supervised Domain Adaptation via Minimax Entropy*
>
>
>
> - The prototype-oriented UDA framework that integrates the prototype idea into a bi-directional transport loss is an original contribution of this paper. It allows us to estimate class proportions, handle the class-imbalance issue, and apply the same model in four different UDA settings (single source, multi-source, class-imbalance, and source-private), achieving state-of-the-art performance while having low computation and memory cost.
>
>
>
> - We agree that using learnable prototypes, which are linear classifiers’ weights, for UDA has already appeared in Saito et al. (ICCV, 2019), which is cited in Section 2.2.1 of our paper when discussing the connections to entropy minimization. We will also cite their paper in the section where we discuss our prototype construction. However, there are at least three key differences. First, they use entropy minimization to align target features to the prototypes whereas we use a bi-directional transport loss. In our ablation study, we make a connection between a one directional (target-to-prototype) transport loss with entropy minimization. This ablation study shows that using this loss alone (similar to entropy minimization) leads to worse performance. Second, Saito et al. (ICCV, 2019) use labeled target examples to construct prototypes, which are not available in our setting. Finally, the proposed prototype-oriented framework allows us to naturally estimate class proportions to handle class imbalance, a problem that is not addressed in Saito et al. (ICCV, 2019).
>
>
>
>
>
>
> >*However, I have one concern related to the quality of this paper. Comparison with proposed baselines seems to be insufficient. Specifically, the proposed method needs to be compared with other prototype-based alignment methods.*
>
>
>
>
> - #### Comparison with CAN (Kang, CVPR 2019):
>
>
>
>      - We will add CAN into comparison for the single-source setting: CAN achieves 90.6% in Office-31 and 65.7% in Office-Home (the result of CAN on Office-Home is quoted from Li et al. (CVPR, 2020)). By comparison, the proposed PCT achieves 90.0% in Office-31 and 71.8% in Office-Home. We note that CAN utilizes source samples to initialize K-means to cluster target samples in each iteration, making it much more computationally expensive and unsuitable for the source-private setting. We also note that different from previous UDA algorithms including ours that typically use domain-agnostic batch normalization, CAN uses domain-specific batch normalization, which is shown to be helpful in boosting its performance.
>
>
>
>
>
>
>
> - #### Comparison with Prototypical Cross-Domain (Yue, CVPR, 2021):
>
>
>
>     - Our performance is better than theirs while being more computationally efficient. They use memory banks and K-means in each iteration to obtain prototypes whereas we simply use the classifier's weight to construct prototypes, saving computation time significantly.
>
>     - In the single-source setting, our average accuracy is higher by 0.2% (90.0 vs 89.8) in the Office-31 dataset and 1.2% (71.8 vs 70.6) in the Office-Home dataset.
>
>
>
> >*I am curious what kind of limitations that this paper could have when compared with other prototype-based alignment papers.*
>
> - Currently, our method is limited to the closed-set domain adaptation setting. There are other prototype-based methods that can handle unseen categories (Saito et al., NeurIPS 2020). It would be interesting to see whether we can extend the proposed PCT method to handle unseen categories, while maintaining its good performance in the closed-set setting.
>
>   $~$
>
> **References:**
>
>
>
> Kang, G., Jiang, L., Yang, Y., & Hauptmann, A. G. Contrastive adaptation network for unsupervised domain adaptation. CVPR, 2019.
>
>
>
> Li, M., Zhai, Y. M., Luo, Y. W., Ge, P. F., & Ren, C. X. Enhanced transport distance for unsupervised domain adaptation. CVPR, 2020.
>
>
>
> Saito, K., Kim, D., Sclaroff, S., & Saenko, K. Universal domain adaptation through self supervision. arXiv preprint arXiv:2002.07953. NeurIPS, 2020.
>
>
>
>
> Saito, K., Kim, D., Sclaroff, S., Darrell, T., & Saenko, K. Semi-supervised domain adaptation via minimax entropy. ICCV, 2019.
>
>
>
> Yue, X., Zheng, Z., Zhang, S., Gao, Y., Darrell, T., Keutzer, K., & Vincentelli, A. S. Prototypical Cross-domain Self-supervised Learning for Few-shot Unsupervised Domain Adaptation. CVPR, 2021.

---

### Official Review · Reviewer_Z3wr · 2021-07-16

**Rating:** 7
**Confidence:** 4

**Summary:**

This paper presents a prototype-based framework for UDA. The core idea is to employ a probabilistic bi-directional transport framework that aligns target domain samples with the source domain prototypes, where the prototypes are modelled using the classifier weights. A key highlight of the method is that it is applicable in the absence of source domain data, which is highly useful in practice. Benchmarking on three datasets shows the method achieves excellent results.

**Limitations And Societal Impact:**

Adequately addressed.

**Main Review:**

Strengths
========
+ The paper is well organized and written. The approach is well motivated.
+ The source-data-private setting is practical. I especially liked the discussion around the use of classifier weights as prototypes, as compared to taking feature mean.
+ The results look promising. The analysis in Section 4 (and Appendix) is comprehensive. The visualizations in Appendix Section E are interesting.

Weaknesses / Clarifications
=====================
I do not have significant criticism of the paper, however addressing the following could improve the submission:

- Regarding fixed $\mu$: Once the model is trained on the source domain, the prototypes are kept fixed. This would enforce the target domain to acquire the semantics learned by the source domain. I agree that this would lead to a more stable training, however, this restriction may not be suitable in category-shift scenarios (e.g., see [P2]). Could the authors comment on this aspect?

- Comparison with prior works: There are several source-data free approaches in the literature [P1, P2] that have been missed in the related work. In particular, [P2] also uses a prototype based metadata to perform “source-free” adaptation, however recommends using separate latent spaces for source and target domain. IMO, these should be discussed in the related work since ideas such as source-data-private setting, and compact feature clusters are already described in these works.

- L86: “missing classes”: It is unclear what is meant by the term “missing classes” here. Does it imply that the method is applicable under category-shift scenario as well (e.g. open-set domain adaptation).

- Classical DA methods work under the theoretical framework of [5] where the adapted model’s generalization bound depends on three terms (source-domain error, divergence between the domains, a constant that depends on the chosen hypothesis space). Here however, in the source-private-data setting, the source domain error becomes immaterial since the feature extractor is updated solely based on the target domain. It would be good to see some theoretical discussion on why the proposed method works (since the adapted feature extractor would not be ideal for the source domain). An interesting analysis could be analyzing the performance of source domain data on the adapted model.

References:

[P1] Kundu et al., “Universal Source-Free Domain Adaptation”, CVPR 2020

[P2] Kundu et al., “Class-Incremental Domain Adaptation”, ECCV 2020

Minor Comments
=============
- Please fix typos (e.g. Fig. 2 caption, “imabalance”). Avoid single worded lines (e.g. Table 5 caption has the word Resnet50 consuming an entire line).
- Fig. 1b is not very informative/novel. This could be removed and the space obtained could be used to plot visualizations shown in Appendix (esp. Class proportion estimation).


**Time Spent Reviewing:**

8

---

> ### Author Response · Authors · 2021-08-09
> **Point-by-point Response to the comments of Reviewer Z3wr**
>
> Thank you for your comments and insightful questions. Below please find our point-by-point response to your comments.
>
>
>
> >*Please fix typos (e.g. Fig. 2 caption, “imabalance”). Avoid single worded lines (e.g. Table 5 caption has the word Resnet50 consuming an entire line).*
>
>
>
> - Noted. We will fix them in the revised version.
>
>
>
> >*Fig. 1b is not very informative/novel. This could be removed and the space obtained could be used to plot visualizations shown in Appendix (esp. Class proportion estimation).*
>
>
>
> - We will remove Fig 1b according to your suggestion and use the saved space to include the class proportion estimation in the main paper.
>
>
>
> >*Regarding fixed : Once the model is trained on the source domain, the prototypes are kept fixed. This would enforce the target domain to acquire the semantics learned by the source domain. I agree that this would lead to a more stable training, however, this restriction may not be suitable in category-shift scenarios (e.g., see [P2]). Could the authors comment on this aspect?*
>
>
>
> - We assume a shared latent space between the source and target features while Kundu et al. (ECCV, 2020) learn separate latent spaces. In the closed-domain adaptation where the categories are the same, we believe this is a reasonable assumption since the semantics of the features from the source domain should be similar to those from the target domain. Otherwise, it might be impossible to do domain adaptation. If we need to handle additional categories, $\mu$ could still be used to initialize the target prototypes since it encodes information about the distribution of the source data. However, care must be taken since allowing $\mu$ to update without the source data could lead to catastrophic forgetting (Goodfellow, 2013). Thus, we will need to add an extra mechanism to deal with this problem.
>
>
>
> >*Comparison with prior works: There are several source-data free approaches in the literature [P1, P2] that have been missed in the related work. In particular, [P2] also uses a prototype based metadata to perform “source-free” adaptation, however recommends using separate latent spaces for source and target domain. IMO, these should be discussed in the related work since ideas such as source-data-private setting, and compact feature clusters are already described in these works.*
>
>
>
> - Thank you for pointing out the references. We will discuss these works in more detail in the final version of the paper.
>
>
>
> - Both Kundu et al. ( ECCV, 2020; CVPR, 2020) use Gaussian prototypes to model each source class distribution. In addition to the parameters for the mean vectors and covariance matrices for each Gaussian, this approach requires an additional decoder network G to compute the reconstruction loss, increasing the memory footprint. While this strategy has an advantage in the category-shift setting, we believe linear classifier's weights are enough to encode information about the source distribution in the closed-set domain adaptation setting. Additionally, our framework also allows us to naturally estimate class proportions to handle class imbalance, a problem that is not addressed in both papers.
>
>
>
> >*L86: “missing classes”: It is unclear what is meant by the term “missing classes” here. Does it imply that the method is applicable under category-shift scenario as well (e.g. open-set domain adaptation).*
>
>
>
> - When we refer to missing classes here, we mean the classes that are not present in the sampled mini-batch. For example, if we have 100 classes and our mini-batch size is 64, some of the classes will definitely be missing. Matching samples with samples could lead to a potential class mismatch. However, the prototypes, the representatives of all the classes, will be available at every iteration. We will modify this sentence to make it more clear.
>
>
>
>
> >*Classical DA methods work under the theoretical framework of [5] where the adapted model’s generalization bound depends on three terms (source-domain error, divergence between the domains, a constant that depends on the chosen hypothesis space). Here however, in the source-private-data setting, the source domain error becomes immaterial since the feature extractor is updated solely based on the target domain. It would be good to see some theoretical discussion on why the proposed method works (since the adapted feature extractor would not be ideal for the source domain). An interesting analysis could be analyzing the performance of source domain data on the adapted model.*
>
>
>
> - Since we fix $\mu$, which encodes information about the distribution of the source sampes, target samples will be aligned to source samples in the latent space. This will decrease the divergence between domains (the second term in the bound). As expected, the model trained on the source domain is going to perform worse on the source data given the adaptation to the target domain. While the first term (source domain errors) increases due to the adaptation, the second term ( divergence between the domains) decreases. Thus, it is inconclusive whether this bound will become looser or tighter.
>
>
>
> - We provide two tables to show the source errors as well as target errors before and after target domain adaptation (Office-31 dataset).
>
>     - | Source Error | A2D   | A2W   | D2A   | D2W   | W2A   | W2D   |
> | ------------ | ----- | ----- | ----- | ----- | ----- | ----- |
> | Before       | 0.99% | 0.90% | 1.00% | 1.00% | 0.00% | 0.00% |
> | After        | 4.90% | 4.93% | 9.04% | 1.00% | 4.91% | 0.10% |
> | Diff         | 3.91% | 4.03% | 8.04% | 0.00% | 4.91% | 0.10% |
>
>     - | Target Error | A2D      | A2W      | D2A      | D2W     | W2A      | W2D     |
> | ------------ | -------- | -------- | -------- | ------- | -------- | ------- |
> | Before       | 19.28%   | 22.01%   | 40.08%   | 5.03%   | 37.06%   | 1.61%   |
> | After        | 7.80%    | 8.30%    | 26.00%   | 2.10%   | 25.40%   | 0.10%   |
> | Diff         | \-11.48% | \-13.71% | \-14.08% | \-2.93% | \-11.66% | \-1.51% |
>
> $~$
>
> **References:**
>
> Goodfellow, I. J., Mirza, M., Xiao, D., Courville, A., & Bengio, Y.. An empirical investigation of catastrophic forgetting in gradient-based neural networks. arXiv preprint arXiv:1312.6211. 2013.
>
>
>
> Kundu, J. N., Venkat, N., & Babu, R. V. Universal source-free domain adaptation. CVPR, 2020.
>
>
>
> Kundu, J. N., Venkatesh, R. M., Venkat, N., Revanur, A., & Babu, R. V. Class-incremental domain adaptation ECCV, 2020.

---

### Official Review · Reviewer_KNFV · 2021-07-16

**Rating:** 7
**Confidence:** 4

**Summary:**

This paper proposes creating prototypes(p) for the source data and aligning these prototypes with the target(t) data via bidirectional conditional transport measure. They use last linear layer's weights as prototypes, so it saves cost, and their method do no require source data for adaptation. For the alignment, they define bidirectional transport, that has  prototypes to targets part and targets to prototypes part in the objective function. For the former, they add prior prototype distributions that comes from Bayes' rule and approximate via EM algorithm. The latter  corresponds to entropy minimization. They obtain state of art results in several domain adaptation scenarios.

**Limitations And Societal Impact:**

The authors mention hardness of eliminating bias problem in domain adaptation problems and energy cost encountered in large deep learning methods.

**Main Review:**


#### *Originality:*

- Their way of using prototypes (using last layers' weights) and formulation of CE resembles supervised contrastive learning(SupCon)[1]. However, this is just one part of their model, and further formulating the alignment in a two way conditional distributional setting and estimating class priors together is novel as of my knowledge.


- Defining the pointwise cost between a prototype and target as doc product makes the whole t->p cost an entropy minimization term, this idea is simple, but the connection is a nice contribution.


- The related work is almost sufficiently cited in my opinion, however a couple of citations might worth adding[2,3,4]


#### *Quality:*

- The claims are supported well in a couple of different ways:

    - The visualizations over synthetic data (Figure 1, 2 and 3) are well designed and convincing.
    - The experiments are extensive and apparently superior
    - Table 6 is a good ablation study that decreases the doubt about the roles of each loss term or their direct alternatives.
    - The connections to the clustering and optimality are simple but enough support.


 - Did you achieve the results on Figure 1b by training a simple case(like Figure 2) or is it just an abstract scenario in order to visualize your idea?


- How would you extend your algorithm to the cases of unseen classes? Would it work by adding nodes and prototypical weights to the output layer?


- The last layer is actually a direct representative of the source classes, is this allowed within the data-privacy practices?

#### *Clarity:*

- Overall, the paper is written clearly and the flow is good.


- There are several improvements I can suggest, for example the authors mentioned advantages of their method(other than main contributions) throughout the paper(such as imbalance, missing class within a batch, parameter costs, outlier robustness etc.) these arguments could be given in a more compact way and in one place.

#### *Significance:*

- The authors achieves really good results on a variety of tasks (single source, multi source, easier and harder datasets), with an intuitive method, their implementation can draw interest.

- Especially given that utilizing prototypes is attempted a fairly good amount of times in unsupervised domain adaptation, and mostly with an adversarial training support and the results are not as superior as the paper at all, again as of my knowledge. So I believe both their method and results will be useful for the field.





**Time Spent Reviewing:**

3

---

> ### Author Response · Authors · 2021-08-06
> **Titles for the recommended references**
>
> Dear Reviewer KNFV, would you please add the titles for references [1-4] pointed out in your review?

---

> ### Author Response · Authors · 2021-08-09
> **Point-by-point Response to the comments of Reviewer KNFV**
>
> Thank you for your insightful review and for recognizing our novel contributions. Below please find our point-by-point response to your comments.
>
>
>
>  >*The related work is almost sufficiently cited in my opinion, however a couple of citations might worth adding[2,3,4]*
>
>
>
> - Thank you for pointing out four additional references. We will discuss and cite them in the final paper if you could provide their titles.
>
>
> >*Did you achieve the results on Figure 1b by training a simple case(like Figure 2) or is it just an abstract scenario in order to visualize your idea?*
>
>
>
> - Figure 1b is an abstract scenario to help visualize the key idea of the proposed method. It tells the same story as the simulated data set shown in Figure 2 does.
>
>
>
> >*How would you extend your algorithm to the cases of unseen classes? Would it work by adding nodes and prototypical weights to the output layer?*
>
>
>
> - As you pointed out, we could add another node of a prototype weight to model out-of-distribution samples (OOD). Instead of predicting K classes, we will predict K+1 classes where the last class is preserved for OOD samples. This scheme, however, will require sampling OOD data, whose space can be arbitrarily large, to train the classifier. If we want to predict the unseen classes, we will need some labeled samples for the novel classes. In this case, we can introduce new prototypes for these samples.
>
>
>
>
> >*The last layer is actually a direct representative of the source classes, is this allowed within the data-privacy practices?*
>
>
>
>
> - While the weights represent the source samples, to the best of our knowledge, the risk of leaking information about the individual users is at least as low as that in Federated Learning (McMahan, B., Moore, E., Ramage, D., Hampson, S., & y Arcas, B. A.. Communication-efficient learning of deep networks from decentralized data. AISTATS, 2017), where privacy is of utmost concern. In the Federated Learning settings, to learn a machine learning model, the whole model instead of data is sent to clients to help protect user's privacy. Similar to Federated Learning, the proposed algorithm sends only the model rather than the source data to the target domain. Thus, we believe it is suitable in the privacy-sensitive setting.
>
>
>
> >*The authors mentioned advantages of their method(other than main contributions) throughout the paper(such as imbalance, missing class within a batch, parameter costs, outlier robustness etc.) these arguments could be given in a more compact way and in one place.*
>
>
>
> - We hoped to emphasize the general applicability of our method so we mentioned the different applications throughout the paper. We will revise the final paper and make the argument more compact per the suggestion.

---

### Decision · Program_Chairs · 2021-09-27

**Decision:**

Accept (Poster)

**Comment:**

Thanks for your submission to NeurIPS.

This paper had somewhat mixed reviews, with 3 mostly positive and 1 more negative reviewer.  During the rebuttal phase, the negative reviewer maintained his/her position that the paper was not ready for publication, so we did not fully reach consensus on the paper.  However, I appreciated the responses that the authors provided to all the reviews, particularly to the more negative reviewer.  There are some valid concerns here---particularly with regard to the experimental comparisons---but overall I am positive about this paper and am happy to recommend it for acceptance at the conference.  Please try to incorporate as much of the rebuttal material as you can in the final paper, as it will strengthen the paper, as well as making sure to address as many of the reviewer criticisms as possible.